

# Testing the reliability of global surface temperature reconstructions of the last glacial cycle

Jean-Philippe Baudouin[1], Nils Weitzel[1], Maximilian May[2], Lukas Jonkers[3], Andrew M. Dolman[4], and Kira Rehfeld[1,5]

[1]Department of Geosciences, University of Tübingen, Tübingen, Germany
[2]Institute of Environmental Physics, Heidelberg University, Heidelberg, Germany
[3]MARUM Center for Marine Environmental Sciences, University of Bremen, Bremen, Germany
[4]Alfred-Wegener-Institut, Helmholtz-Zentrum für Polar-und Meeresforschung, Potsdam, Germany
[5]Department of Physics, University of Tübingen, Tübingen, Germany

**Correspondence:** Jean-Philippe Baudouin (jean-philippe.baudouin@uni-tuebingen.de)

**Abstract.**

Reconstructing past variations of the global mean surface temperature is used to characterise the Earth system response to perturbations as well as validate Earth system simulations. Reconstructing GMST beyond the instrumental period relies on algorithms aggregating local proxy temperature records. Here, we propose to establish standards for the evaluation of the performance of such reconstruction algorithms. Our framework relies on pseudo-proxy experiments. That is, we test the ability of the algorithm to reconstruct a simulated GMST, using artificially generated proxy data created from the same simulation. We apply the framework to an adapted version of the GMST reconstruction algorithm used in Snyder (2016), and the synthesis of marine proxy records for temperature of the last $130\,\mathrm{kyr}$ from Jonkers et al. (2020). We use an ensemble of 4 transient simulations of the last glacial cycle or the last $25\,\mathrm{kyr}$ for the pseudo-proxy experiments.

We find the algorithm to be able to reconstruct timescales longer than $4\,\mathrm{kyr}$ over the last $25\,\mathrm{kyr}$. However, beyond $40\,\mathrm{kyr}$ BP, age uncertainty limits the algorithm capability to timescales longer than $15\,\mathrm{kyr}$. The main sources of uncertainty are a factor, that rescales near global mean sea surface temperatures to GMST, the proxy measurement, the specific set of record locations, and potential seasonal bias. Increasing the number of records significantly reduces all sources of uncertainty but the scaling. We also show that a trade-off exists between the inclusion of a large number of records, which reduces the uncertainty on long time scales, and of only records with low age uncertainty, high accumulation rate, and high resolution, which improves the reconstruction of the short timescales.

Finally, the method and the quantitative results presented here can serve as a basis for future evaluations of reconstructions. We also suggest future avenues to improve reconstruction algorithms and discuss the key limitations arising from the proxy data properties.





## 1 Introduction

The global mean surface temperature (GMST) is a fundamental quantity to describe climate change. It is a major component of the Earth's energy balance and describes the Earth system response to perturbations. Furthermore, the GMST is one of the variables characterising Earth's habitability with consequences on ecosystems. Hence, GMST is the main target measure of international efforts to limit anthropogenic global warming (UNFCCC, 2015). Past evolution of GMST is used to evaluate

Earth System Models (ESMs, e.g. PAGES 2k Consortium, 2019; Brierley et al., 2020; Kageyama et al., 2021; Lunt et al., 2021) and to estimate the equilibrium climate sensitivity, i.e. the response of GMST to a doubling of atmospheric $CO_2$ concentrations (Snyder, 2016; Friedrich and Timmermann, 2020).

In order to project climate change for the end of the century and beyond, comprehensive ESMs are being developed. Some of these ESMs include an atmosphere-ocean general circulation model along with modules for ice sheets (Nowicki et al., 2016;

Ziemen et al., 2019; Muntjewerf et al., 2020), iceberg melting (Rackow et al., 2017; Erokhina, 2020), dynamic vegetation (Song et al., 2021), and the carbon cycle (Brovkin et al., 2012; Arora et al., 2020; Kleinen et al., 2021). During the Last Glacial Cycle (LGC, the last $130\,\mathrm{kyr}$, which includes the Last Interglacial (LIG), the last glacial period, and the Holocene), large changes in these Earth system components occurred in response to radiative forcing changes of comparable magnitude to those projected in future scenarios. Therefore, the LGC is an important evaluation period for ESMs.

Reconstructing past GMST relies on proxy records extracted for example from sediments or ice. Many methods for reconstructing the GMST use a bottom-up approach, where a large number of local proxy temperature records are combined to compute area-averaged temperature (e.g. Shakun et al., 2012; Snyder, 2016; Friedrich and Timmermann, 2020; Osman et al., 2021; Clark et al., 2024). This approach aims at suppressing noise and regional influences in these time series but remains sensitive to the spatial and temporal coverage of the data. It also relies on a homogeneous dataset for consistency between

the records. For the LGC, a new database has recently been published: the *PalMod 130k marine palaeoclimate data synthesis* (Jonkers et al., 2020, hereafter J20). This database provides numerous proxy records of near-surface sea temperature from marine sediments. The records are subject to uncertainties from dating and proxy-to-temperature relationship (Bradley, 2014). In particular, several processes influence the relationship between the Sea Surface Temperature (SST) and the measured proxy value, such as other variables than SST, bioturbation, and sampling biases. These limitations lead to uncertainties that need to

be accounted for in any GMST reconstruction. The J20 database is metadata-rich, which enables the characterisation of these uncertainties.

Compared to the Common Era or the Holocene, for which many methods have been developed and reconstructions computed (see for example Consortium, 2017; PAGES 2k Consortium, 2019; Kaufman et al., 2020), the LGC presents the challenge of a non-stationary climate state and sparser records. Snyder (2016, hereafter S16) developed an algorithm to overcome this

challenge and provides a GMST reconstruction of the Pleistocene to investigate glacial cycles. It is a robust algorithm which, compared to others, accounts for data sparsity, quantifies uncertainties and has a limited reliance on model output. Therefore, this algorithm is a good basis for developing evaluation standards and assessing reconstruction algorithm performances and



limitations. In particular, we note several open questions regarding the performances of algorithms such as S16 to reconstruct the GMST during and beyond the LGC:

– How reliable are GMST reconstructions of the LGC based on SST proxies?

    – Does the non-uniform spatiotemporal distributions of proxy samples lead to biased GMST reconstructions?

    – What are the contributions of the various sources of uncertainty?

    – What is the shortest timescale on which amplitudes and timing of GMST variations can be accurately reconstructed?

    – What are the sources of the loss of accuracy on short timescales?

– Which limiting factors should be prioritised for improving the GMST reconstruction quality (uncertainty, resolution)?

We study these questions using pseudo-proxy experiments (PPEs) based on transient climate simulations. PPEs enable the test of algorithms in controlled, idealised environments, where the underlying truth is known (i.e. the simulation outputs). They rely on pseudo-proxies that are synthetic time series imitating proxy records, based on the spatiotemporal climate state of the simulation. The computation of pseudo-proxies is performed using a proxy system model (*sedproxy*, Dolman and Laepple,

2018), which simulates the processes from the fixation of the climate-dependent quantity to the measurement of the proxy in the lab, including the entailed uncertainties. The use of metadata from the J20 database to compute pseudo-proxies provides a realistic example for GMST reconstruction. By applying the reconstruction algorithm to various sets of pseudo-proxy data, we can evaluate the performance of the reconstruction method, as well as the quality of the potential LGC reconstruction based on the J20 dataset.

In the following, we first describe the J20 database and the transient climate simulations employed to compute the pseudo-proxies. Then, we present the GMST reconstruction algorithm that we evaluate and the design of the PPEs. The results from the PPEs follow it. Finally, we discuss the performance of the algorithm when applied to the J20 database, and future avenues to improve GMST reconstructions.



## 2 Data

### 2.1 Temperature reconstructions

Our study relies on one of the largest synthesis of marine proxy records for temperature of the LGC available: the extended *PalMod 130k marine palaeoclimate data synthesis*, beta version 2.0.0 (Jonkers et al., 2020, hereafter J20). This dataset is a multi-proxy compilation of globally distributed marine proxy records spanning the last $130\,\mathrm{kyr}$ and was developed within the *PalMod* initiative (https://www.palmod.de; Latif et al., 2016; Fieg et al., 2023) as a comprehensive reference for transient climate simulations. We prefer this database over the one assembled for S16 as it enables a more comprehensive uncertainty assessment, needed to develop our evaluation standards. In particular, it includes richer metadata and a unified framework for the age models. The age-depth models are constrained using a blend of radiocarbon dates (between 0 and $\sim 40\,\mathrm{kyr}$ BP) and $\delta^{18}$O benthic stratigraphy based on Lisiecki and Stern (2016). Chronological uncertainty is assessed using the Bayesian framework BACON (Blaauw and Christen, 2011). Specifically, the database contains 1000 age ensemble members for each sediment core.

We selected all proxy records for SST located within the latitudes $60°$N and $60°$S, following S16, with at least 10 values. This leaves us with 265 time series at 189 locations. These records rely on temperature dependencies of the chemical composition, chemical product, and the temperature preferences of various organisms living near the sea surface. Here, this includes alkenone indices ($U_{37}^{k}$, $U_{37}^{k'}$, 89 time series, Prahl et al., 1988), a lipid-based index (TEX$_{86}$, 6, Schouten et al., 2002), the long chain diol index (LDI, 4, Rampen et al., 2012), $Mg/Ca$ ratios in planktonic foraminifera (103, Nürnberg et al., 1996), and microfossil assemblages (planktonic foraminifera, radiolaria, and diatoms, 63, Imbrie, 1971). In addition, we only kept the time series for which the representative season was indicated as annual (105 time series) or unspecified (112). In some cases, assemblages from the same core have been used to estimate winter and summer temperatures. If the timestep coincides, we average the two to form 48 "pseudo-annual" time series.

Record metadata show some distinctive features (Fig.1 and 2). First, record locations tend to better cover the Atlantic than the Indian and Pacific oceans, and to be close to the coast (Fig.1-A). Second, the number of proxy records is maximum for the deglaciation (>200), while it is relatively constant between 40 and $120\,\mathrm{kyr}$ BP (82, Fig. 2-A). Yet, substantial data gaps occur in the Southern Hemisphere for periods prior to the LGM (Fig.1-B). Third, the lowest age uncertainty is achieved in the last $30\,\mathrm{kyr}$ (<3 kyr), where radiocarbon dating is reliable and frequently employed. For the rest of the LGC, median age uncertainty is slightly below $10\,\mathrm{kyr}$. Finally, the highest temporal resolution and the highest accumulation rate are also found in the last $30\,\mathrm{kyr}$ (< 1 kyr). Note that a high accumulation rate leads to a bioturbated layer covering a shorter period of time.

### 2.2 Climate model output

To compute the pseudo proxy time series used in the PPEs, we use transient simulation outputs from four different climate models: FAMOUS, LOVECLIM, CESM, and MPI-ESM. This model-model approach is used to test the robustness of the result.



FAMOUS, the Fast Meteorological Office and UK Universities Simulator, is a version of the Hadley Centre Coupled Model version 3 (HadCM3, Gordon et al., 2000) AOGCM with reduced resolution (Smith et al., 2008). The simulation used here is the version ALL-5G produced in the QUEST project, and covers the last 120 kyr (Smith and Gregory, 2012). The simulation is forced with transiently changing orbital parameters (Berger, 1978), greenhouse gas concentrations (Spahni et al., 2005; Lüthi

et al., 2008), and Northern Hemisphere ice sheet extents and topographies from the ICE-5G v1.2 dataset (Peltier, 2004)[1]. The time-varying external boundary conditions are accelerated by a factor of ten in the simulation, such that the last 120 kyr are effectively simulated in 12 kyr in the model (Smith and Gregory, 2012).

LOVECLIM (LOch–Vecode-Ecbilt-CLio-agIsm Model) is an Earth system model of intermediate complexity (Goosse et al., 2010), that is, with coarser spatial resolution, and simpler representation of physical processes than in general circulation mod-

els. The simulation used here (Loveclim 800k, Timmermann and Friedrich, 2016) uses version 1.1 of LOVECLIM (Goosse et al., 2007), with only the atmospheric, oceanic, and vegetation modules activated. Greenhouse gas concentrations are prescribed following the measurements from the EPICA DOME C ice core (Lüthi et al., 2008), and the evolution of ice sheets (surface elevation and albedo, land-sea mask) are prescribed following a simulation with the CLIMBER-2 model (Ganopolski and Calov, 2011). Finally, orbital configurations are derived from Berger (1978). Similarly to FAMOUS, the external forcings

are accelerated by a factor of five.

Several simulations of the last 26 kyr, using MPI-ESM 1.2 (Mauritsen et al., 2019), have been published in Kapsch et al. (2022). We select two simulations from this paper. The first simulation (called GLAC1D-P2 in Kapsch et al. (2022), hereafter MPI-G1D) is forced using the GLAC-1D ice sheet reconstruction (Tarasov et al., 2012; Briggs et al., 2014). By contrast, the second simulation (ICE6G-P2 in Kapsch et al. (2022), hereafter MPI-I6G) uses ICE-6G (Peltier et al., 2015). Forcings are

otherwise identical (Köhler et al., 2017; Berger, 1978, for greenhouse gases and insolation respectively). These simulations also include meltwater flux forcing from ice sheet melt through a dynamical routing.

Finally, we use a simulation of the last three million years (Yun et al., 2023), based on CESM 1.2. (Hurrell et al., 2013). For our period of interest, it uses as boundary conditions LGM land-sea mask, greenhouse gas concentrations from EPICA DOME C ice core (Lüthi et al., 2008), orbital configurations from Berger (1978), and the ice sheet elevation and albedo from

a CLIMBER-2 simulation (Willeit et al., 2019). The time-varying boundary conditions are accelerated by a factor of five. Open-access data availability is limited to millennial averages, which restricts the use of this simulation in our analysis.

---

[1]The land-sea mask is that of the pre-industrial condition in this simulation, so the ice sheet topography is only applied to the pre-industrial land surface





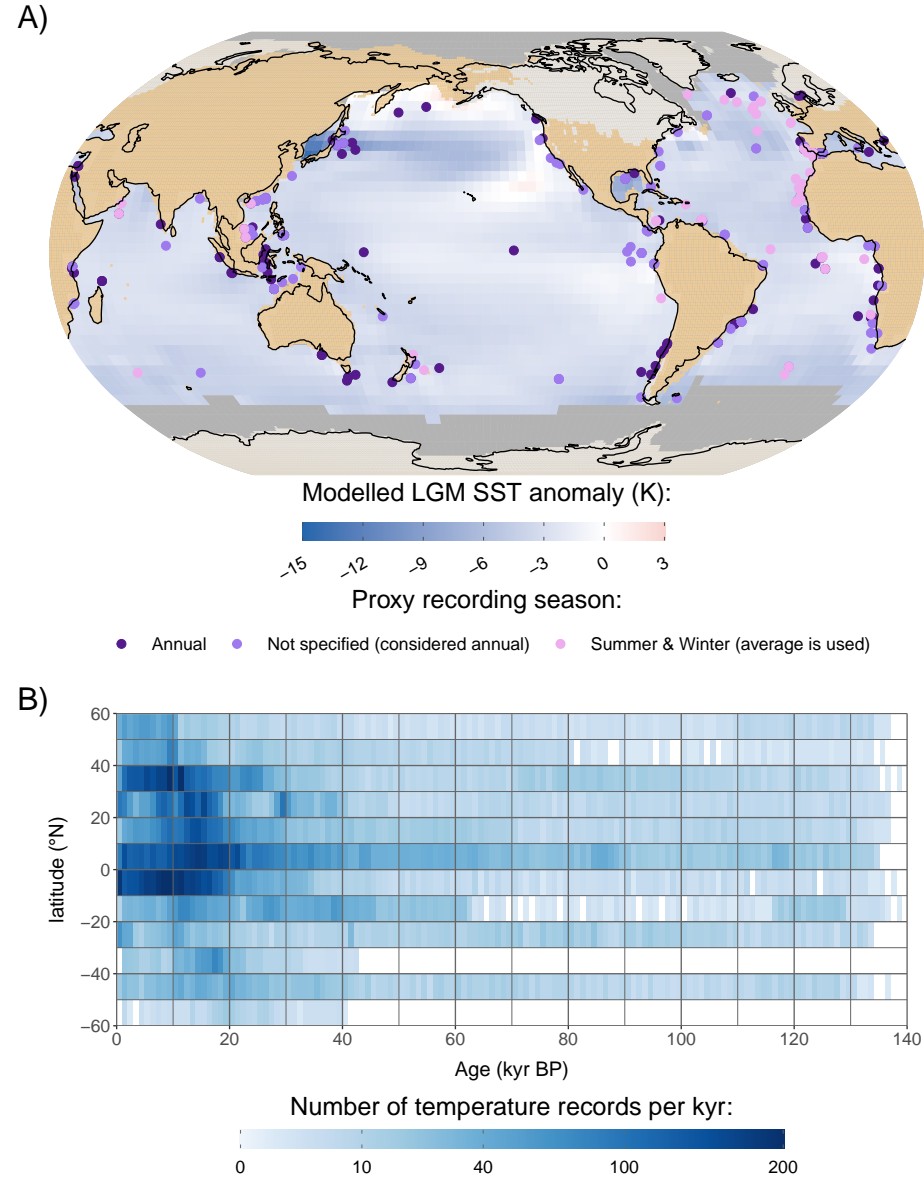

**Figure 1.** Spatial and temporal distribution of the selected temperature proxy records. Panel A shows the locations of inferred SST time series between $60°$ N and $60°$ S available in version 2.0 of the *PalMod 130k marine palaeoclimate data synthesis* used for this study. Colours indicate whether the recording seasons of the time series are labelled as annual (violet), not indicated (purple), or whether both summer and winter time series are available (lilac). The background shows, over the ocean, the LGM ($19$-$23$ kyr BP) anomaly of sea surface temperature (SST) from MPI-I6G as an example (see sect. 2.2). Ice-free land (brown) and ice sheets (light grey) are taken from ICE-6G (Argus et al., 2014; Peltier et al., 2015). Darker grey in the ocean corresponds to sea-ice-covered areas in the simulation. Panel B represents the latitudinal and temporal distribution of the SST proxy time series from the above panel. The number of proxy data points is given per latitudinal bands of $10°$ and 1 kyr.

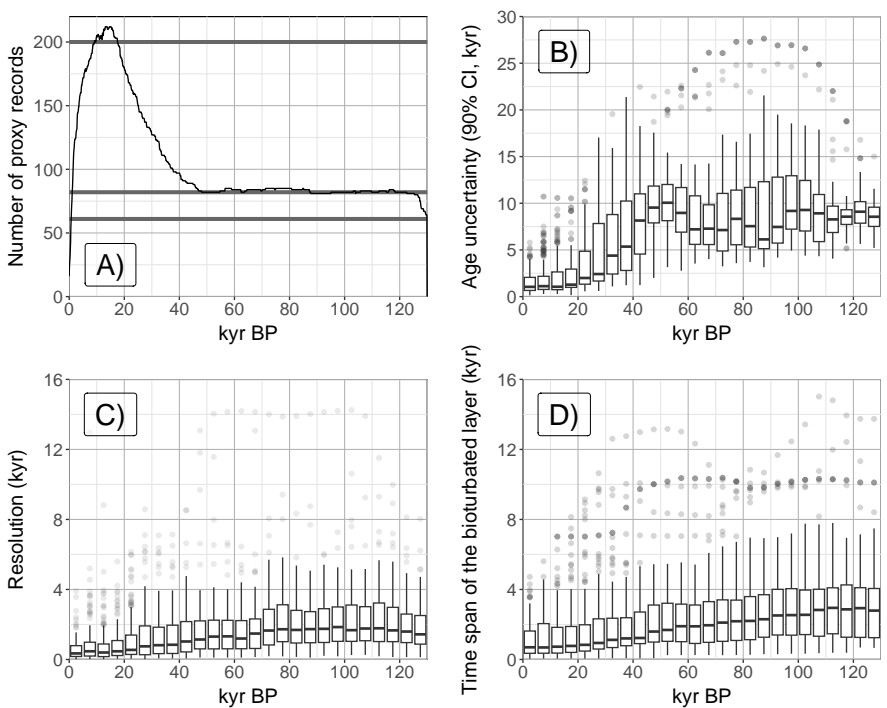

**Figure 2.** Characteristics of the proxy records in the J20 database. In panel A, the number of records is computed using the entire time span of the time series. The grey horizontal lines correspond to the values 61, 82, and 200, used in the pseudo-proxy experiments (cf. Tab. 2). In panels B, C and D, the investigated characteristic is averaged over a 5 kyr bin for each timeseries, so that the box plots represent the distribution over the ensemble of records. The bioturbated layer in panel D is computed as the bioturbated layer width (fixed at 10 cm) divided by the accumulation rate, so that the layer width can be interpreted as a period of time.



## 3  Methods

We first present the GMST reconstruction algorithm, which is an adaptation of S16 (sect. 3.1). Then, we describe the design of
PPEs to evaluate the reconstruction quality (sect. 3.2).

### 3.1  Global mean surface temperature reconstruction

### 3.1.1  The S16 algorithm

The original S16 algorithm reconstructs GMST over the Pleistocene following four main steps.

First, all the irregular raw time series are brought to a common, equidistant ($1\,\mathrm{kyr}$) time axis. Then, the time series are
smoothed using a Gaussian kernel which represents the effect of age uncertainty (a constant $10\,\mathrm{kyr}$, as the 95% interval).
The algorithm also estimates the signal's standard deviation arising from age, measurement, and calibration uncertainty. The
measurement and calibration uncertainty is defined as a standard deviation of $1.5\,\mathrm{K}$, a value similar to other reconstruction
studies (Shakun et al., 2012; Tierney et al., 2020). The SST anomalies are finally computed with respect to the late Holocene
(0 to $5\,\mathrm{kyr}$ BP).

Second, the considered domain between $60°S$ and $60°N$ is subdivided into latitudinal bands (with 9 different configura-
tions), used to cluster the records. The objective is to calculate SST anomalies for each band. Therefore, this latitudinal sub-
division is based on the assumption that temperature anomalies are approximately homogeneous within each band (Rohling
et al., 2012). This method is a trade-off to still account for the heterogeneity of the record locations, despite the sparsity of the
available SST records. At this stage, a Monte Carlo approach is used to propagate the signal uncertainty from each record to
the zonal band averages. Specifically, independent realisations of the signal uncertainty are used for each record to compute
one thousand zonal band averages, for each of the 9 band configurations. Then, n timeseries are randomly selected from all
the time series available (the number of records times the one thousand realisations of the previous sampling), where n is the
number of records. That way, the algorithm can evaluate the uncertainty related to the selection of records, and in particular to
their location.

Third, the zonal band averages are aggregated to derive a $60°S$ to $60°N$ SST anomaly.

Fourth, the mean SST anomaly is scaled to a GMST anomaly. Similar to other studies (Bereiter et al., 2018; Friedrich and
Timmermann, 2020), a linear scaling coefficient is used to account for the stronger cooling of polar and terrestrial regions
compared to the spatial mean SST change between $60°S$ and $60°N$. The coefficient is derived from PMIP2 and PMIP3 LGM
and pre-industrial simulations (Braconnot et al., 2007, 2012). A Monte Carlo approach is again used to quantify the uncertainty:
five thousand different realisations of the scaling factor (defined as a Gaussian distribution with a mean of 1.9 and a standard
deviation of 0.2) are applied to randomly selected realisations of the mean SST anomaly.





### 3.1.2 Adaptation of S16 algorithm

We decide to adapt the S16 algorithm to make full use of the J20 dataset, and improve the uncertainty quantification by the algorithm.

First, we apply the measurement and calibration noise before the interpolation of the raw time series. That way, the un-
certainty depends on the number of measurements performed: averaged over the same time span, a high-resolution record should have a reduced uncertainty compared to a low-resolution one. We consider a thousand different samples of this noise to quantify the uncertainty.

Second, we make use of the age ensembles available for each record in the J20 database: each time series is interpolated using one of the 1000 age ensemble members. A different realisation of the measurement and calibration noise is used for
each of these time series. This method lends itself to the Monte Carlo approach used later in the S16 algorithm and avoids any assumption on the distribution of the age uncertainty. We also use a centennial timestep for the interpolation, so that the higher frequency variability of the reconstructed GMST can be investigated.

Third, 96 records from the J20 database do not cover the reference period 0-5 kyr BP, from which the SST anomaly is computed. Including them increases the number of available records for the period 40-120 kyr BP by 60%. Therefore, we
implement a zonal iterative offsetting procedure. We assume that the SST anomalies are homogeneous enough within each latitudinal band, so that the offset needed to compute the SST anomaly of one record can be derived from the others. In each of the zonal bands, the time series are adjusted iteratively, with respect to all previously adjusted time series, according to the following steps:

1. Preliminary step: all records in the considered zonal band are sorted by the age of their most recent data point, starting with the closest to the present.

2. First offsetting: the temperature anomaly of the record including the most recent age is computed with respect to the most recent 5 kyr of that same record.

3. Iterative offsetting: we now consider the interval consisting of the most recent 5 kyr of the next non-adjusted record with the most recent age. We compute the mean of all the already adjusted time series in the zonal band over this interval. The offset of the new record is then calculated such that this mean equals the average temperature anomaly of the new record over the same interval.

Fourth, all latitudinal bands are averaged using an area-weighted mean where the weight is proportional to the oceanic surface, instead of the entire surface of the band in S16. For this, we use the ICE-6G reconstruction (Peltier et al., 2015) of the LGM for the entire time period, assuming that changes in oceanic surfaces in each band are small enough compared to other
uncertainty sources to be neglected.

Finally, as in S16, the resulting Monte Carlo realisations are used to compute summary statistics of the GMST reconstruction, namely mean and 95% confidence interval. However, in our case, each GMST time series does not span exactly the same time



period, because they are based on different sets of age models. Therefore, the statistics are computed only when values from at least half of the realisations are available.

## 195 3.2 Pseudo-proxy experiments

We evaluate the algorithm using pseudo-proxy experiments (PPEs). Pseudo-proxies are artificially generated proxy data created from a reference climate evolution, in our case, temperature outputs from transient simulations. Thus, we apply the GMST reconstruction algorithm to these pseudo-proxies to analyse how well the reference GMST evolution, i.e. the GMST of the transient simulations, is recovered by the reconstruction algorithm. In this idealised setting, influences of individual aspects of

the algorithm or characteristics of the data can be studied using sensitivity experiments. In the following, we first describe the strategy to construct pseudo-proxies before presenting the design of the PPEs we perform.

### 3.2.1 Construction of pseudo-proxies with *sedproxy*

Pseudo-proxies are created by simulating the processes that contribute to the recording of a climate signal in an archive. Here, we use the *R* package *Sedproxy* to simulate these processes for marine sediment records (Dolman and Laepple, 2018, 2023).

More specifically, the package provides a forward model that converts an ocean temperature time series with monthly resolution, into pseudo-proxy time series with the desired resolution. Table 1 summarises the processes simulated for each proxy type and the associated metadata. The algorithm implemented in *sedproxy* consists of four steps.

First, the input temperature data is converted to the relevant proxy unit using well-established calibrations (forward model). This forward model represents the dependency of a process to ocean temperature (the sensor), such as the calcite formation in

foraminiferal tests, which uses both Ca or Mg ions, in a proportion depending on the temperature (Nürnberg et al., 1996). This model corresponds to the inverse of the calibration function, used to infer temperature from proxy measurements. This step is needed to consider the structure of the calibration error. Note that the calibration for $TEX_{86}$ and $LDI$ is not yet implemented in *sedproxy*. Given that it concerns only ten records, the impact on our analysis should not be significant. In addition, calibration methods used for assemblages in J20 are heterogeneous, and no established forward models, as needed in this first step, exist.

Hence, no conversion is performed for these 63 records either. This limits insight into the error structure, but it cannot be overcome with our current framework.

Second, for each time point (in fact, sediment depth) for which the proxy record is to be modelled, the algorithm computes monthly weights. These weights represent the chance for the sensor to have recorded the temperature of that specific month and year. The weights combine effects from seasonal preferences of species, bioturbation, and sediment thickness of the sample

(for an exhaustive description, see Dolman and Laepple, 2018). In *sedproxy*, seasonal preferences are only implemented for $Mg/Ca$ ratios, although models for alkenones do exist (Tierney and Tingley, 2018). These preferences are based on the recording species indicated in J20, and computed in *sedproxy* using the FAME v1.0 model (Roche et al., 2018). For 16 of the records, no seasonal preference is applied, as no growth model is provided for the associated species (namely N. incompta, G. inflata, G. truncatulinoides, P. obliquiloculata and G. crassaformis). Bioturbation is modelled as a low-pass filter. It supposes

a constant bioturbated layer width, which is fixed at $10\,\mathrm{cm}$ in our study (default value, cf. Zuhr et al., 2022). It also requires





sedimentation rate as input, which we derive from the depth and mean age available in J20. The impact of the sediment thickness of the sample is also modelled at this stage. In our study, it is computed from the difference between two sample depths, but not higher than $2\,\mathrm{cm}$. Finally, for $Mg/Ca$ ratios, *sedproxy* samples 30 pseudo-proxy values whose temporal distribution follows that given by the monthly weights. This sampling corresponds to the number of foraminifera needed to produce a measurement
(default value, Dolman and Laepple, 2018). *sedproxy* also adds to these values an error corresponding to the inter-individual variability (standard deviation of $2\,\mathrm{K}$, Dolman and Laepple, 2018), and eventually takes the average over the 30 samples. This sampling process is irrelevant for both $U_{37}^{K'}$ and $TEX_{86}$: the sample size is considered infinite, and the pseudo-proxy values are based on a weighted mean using the monthly weights. For assemblages, this sampling would be relevant but what is implemented in *sedproxy* is too simple to represent it and dependent on a forward model (cf. previous paragraph), and therefore
not considered.

The third stage of the algorithm consists of applying a random Gaussian noise depending on the measurement error. This one is set to $0.26\,\mathrm{K}$ and $0.23\,\mathrm{K}$ for $Mg/Ca$ ratios and $U_{37}^{K'}$ respectively (Dolman and Laepple, 2018). Unfortunately, no standard measurement errors are available for $TEX_{86}$ and assemblages, so we arbitrarily set it to $1\,\mathrm{K}$. This large uncertainty aims to quantify the impact of not using forward models for the temperature to proxy calibration.
The fourth step of the algorithm consists of converting the time series from proxy unit back to temperature. Calibration uncertainty, which arises from using empirical relationships inferred from imperfect data, is only considered at this step. This is done by sampling the calibration parameters using a Gaussian distribution.

In addition to, and before applying *sedproxy*, we add a step to account for age uncertainty. Specifically, we randomly select one age member among the one thousand available, so that *sedproxy* computes the pseudo-proxy values at these specific time
points. The resulting time series is then associated with all the age ensemble members available.

Finally, the input data for *sedproxy* corresponds to the temperature of the oceanic grid point nearest to the proxy location. For LOVECLIM, MPII6G and MPIG1D, we use the topmost level of ocean temperature. In addition, we use air temperature for FAMOUS and surface temperature for CESM, with a screen for land mass and values below -2°C, so that the seasonality can be investigated.

**3.2.2 Specification of the pseudo-proxy experiments**

Different sets of PPEs are designed, first to evaluate the performance of the algorithm, and then to test the influence of various data properties on the reconstruction. The characteristics of the pseudo-proxies computed for each PPE are summarised in table 2. In the following, we further describe the method to compute the pseudo-proxies and discuss the rationale of each experiment.

The *Full PP* experiment is used to evaluate the performance of the algorithm under normal conditions. The pseudo-proxies
are computed as described in sect. 3.2.1. The computation of the pseudo-proxies is performed ten times to test the robustness of our results to the random elements in the methodology (measurement, calibration, and age uncertainty when applying *sedproxy*). The individual contributions of the five sources of uncertainty quantified by the algorithm are also determined. We indicate in the following in parentheses how each uncertainty quantification is switched off: the measurement and calibration noise (no noise is added), the age model ensemble (only the mean age is considered), the selection of records (all time series





**Table 1.** Processes simulated in *sedproxy* for each type of proxies. The values are those suggested in Dolman and Laepple (2018).

| Proxy type | $Mg/Ca$ | $U_{37}^{K'}$ | $TEX_{86}$ | LDI | Assemblages |
|---|---|---|---|---|---|
| Number of records | 103 | 89 | 6 | 4 | 63 |
| Transfer function | Anand et al. (2003) | Müller et al. (1998) | None used | None used | None used |
| Seasonal bias | Yes (for supported species) | Not considered | | | |
| Bioturbation | 10 cm | | | | |
| Sediment layer width | distance between each measurement (if <2 cm) | | | | |
| Sampling size | 30 | Infinite sample size | | | |
| Inter-individual variations | 2 | Not defined | | | |
| Measurement error (K) | 0.26 | 0.23 | 1 | 1 | 1 |

are used), the latitudinal band configurations (only one configuration is considered: 60°N - 40° - 20°N - 0° - 20°S - 40°S - 60°S), and the scaling factor (no uncertainty is applied).

The *measurement noise* experiment only includes the measurement uncertainty from *sedproxy* (along with the calibration uncertainty), and its estimate from the algorithm. The experiment is used to compare the representation of measurement noise in the algorithm with the one introduced in the pseudo-proxy, as well as to quantify the effect of the latter on the signal quality. In order to investigate this effect, we perform 30 iterations of the PPE.

The experiment *SST at proxy locations* is designed to evaluate the ability of the algorithm to recover the 60°S-60°N mean SST signal, given the number and locations of the proxy data. None of the processes modelled by *sedproxy* is included, that is, the GMST algorithm is directly provided with the modelled SST timeseries at the grid point nearest to the proxy location.

*Full PP at random locations* corresponds to a series of PPEs investigating the influence of the number of proxy records on the uncertainty. Four numbers are tested: 61 is the number of records selected in S16, 82 is the average number of records during MIS 3 to 5 in our dataset (Fig. 2-A), 200 is the minimal number of records during the deglaciation, and 400 is an example in case a much larger dataset could be obtained. Oceanic locations are randomly generated for each pseudo-proxy. To each of these locations, we assign metadata (proxy type and species), that corresponds to a random record in J20 located within 5° of latitude. To construct age models, we start by selecting one hundred age ensemble members from each of the 44 (for the MPI simulations) or 24 (for FAMOUS and LOVECLIM) sediment cores that fully span respectively the deglaciation ($2-25\,\mathrm{kyr}$) and the LGC ($5-130\,\mathrm{kyr}$). The age models are then interpolated so that their mean resolution is a hundred years. This creates 44 or 24 age ensembles that are randomly assigned to a location. Finally, these metadata are used to compute pseudo-proxy time series with *sedproxy*. Effects from bioturbation and sediment thickness are removed, as they do not impact the uncertainty estimates of the algorithm. This procedure is iterated 100 times for each number of pseudo-proxy records tested, so that an uncertainty range can be drawn from the ensemble.

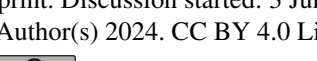



| Pseudo-Proxy Experiments | Simulations | Proxy characteristics | | | | Seasonality | *Sedproxy* Bioturbation | Measurements | Uncertainties quantified in the GMST algorithm | Members | Figures |
| | | Number | Locations | Age resolution | Age uncertainty | | | | | | |
| --- | --- | --- | --- | --- | --- | --- | --- | --- | --- | --- | --- |
| Full PP | MPIs, FAMOUS, LOVECLIM | 225 - 265 | J20 | J20 | J20 | only *Mg/Ca* and not LOVECLIM | J20 | Yes | First all, then each separately | 10 | 3, 4,7,8 |
| Measurement noise | MPIs, FAMOUS, LOVECLIM | 225 - 265 | J20 | 100 yr | - | - | - | Yes | Measurement | 30 | 8 |
| SST at proxy locations | MPIs, FAMOUS, LOVECLIM, CESM | 225 - 265 | J20 | 100 yr | - | - | - | - | resampling and latitudinal bands | 1 | 5-A |
| Full PP at random locations | MPIs, FAMOUS, LOVECLIM | 61, 82, 200, 400 | Random | 100 yr | J20* | only *Mg/Ca* and not LOVECLIM | - | Yes | First all, then each separately | 100 | - |
| No offsetting | MPIs, FAMOUS, LOVECLIM | 225 - 265 | J20 | J20 | J20 | only *Mg/Ca* and not LOVECLIM | J20 | Yes | - | 10 | 6-A |
| Warmest month | MPIs, FAMOUS, LOVECLIM, CESM | 225 - 265 | J20 | 100 yr | - | Warmest month | - | - | - | 1 | 6-B |
| Bioturbation | MPIs, FAMOUS, LOVECLIM | 225 - 265 | J20 | 100 yr | - | - | J20* | - | - | 1 | 8 |
| Age uncertainty | MPIs, FAMOUS, LOVECLIM | 225 - 265 | J20 | 100 yr | J20* | - | - | - | - | 1 | 8 |
| Age uncertainty and bioturbation | MPIs, FAMOUS, LOVECLIM | 225 - 265 | J20 | 100 yr | J20* | - | J20* | - | - | 1 | 8 |
| Age resolution | MPIs, FAMOUS, LOVECLIM | 225 - 265 | J20 | J20* | - | - | - | - | - | 100 | 8 |
| Measurement noise at age resolution | MPIs, FAMOUS, LOVECLIM | 225 - 265 | J20 | J20* | - | - | - | Yes | - | 100 | 8 |

**Table 2.** Characteristics of the PPEs. The first set is used to analyse the uncertainty sources. The second set, with the addition of *Full PP at random locations*, is used to investigate the representation of the time continuum. Apart from *Full PP at random locations*, the number of timeseries considered depends on the simulation time span: 225 (MPI), 263 (FAMOUS), 265 (LOVECLIM, CESM). J20 corresponds to the use of metadata from J20. J20* refers to special cases described in the text. Some PPEs are computed several times to form an ensemble. It allows the sampling of measurement noise, random location, or reconstructed age ensemble when applied. The figures where the corresponding PPE is used are indicated in the last column. The results of *Full PP at random locations* are only discussed in the text.



The *no offsetting* experiment is used to evaluate the impact of the offsetting procedure: this one is removed from the algorithm and the correct temperature anomaly of the simulation at each location is used instead. The experiment is otherwise identical to the *Full PP* experiment. The *warmest month* experiment is used to estimate an upper bound of seasonality bias on the proxy records. The pseudo-proxies are computed by only considering the warmest month within each year.

The next three experiments evaluate the individual impact on the signal of bioturbation, age uncertainty and age resolution respectively, using a simple and idealised setup. In the *bioturbation* experiment, the accumulation rate is identical at each location and is equal to the average across the entire J20 dataset (but still time-dependent, cf. Fig. 2-D). Note that we consider bioturbation together with the sediment thickness, as the two are intertwined in *sedproxy*. For the *age uncertainty* experiment, we use the same 44 or 24 age ensembles computed for the experiment *Full PP at random locations* to consider age uncertainty

independently from other parameters. We consider this selection of age ensembles to closely represent the age uncertainty as a function of time for the entire proxy dataset (as given in Fig. 2-B). Finally, for the *age resolution* experiment, we construct age models for the entire time period which resemble the original distribution of temporal resolutions across time (Fig 2-C). The experiments *Age uncertainty and bioturbation* and *Measurement noise at age resolution* investigate combined effects.

Finally, note that CESM data has a too low temporal resolution to be able to consider the smoothing factors applied by

*sedproxy*, and that LOVECLIM data doesn't enable the investigation of seasonality.





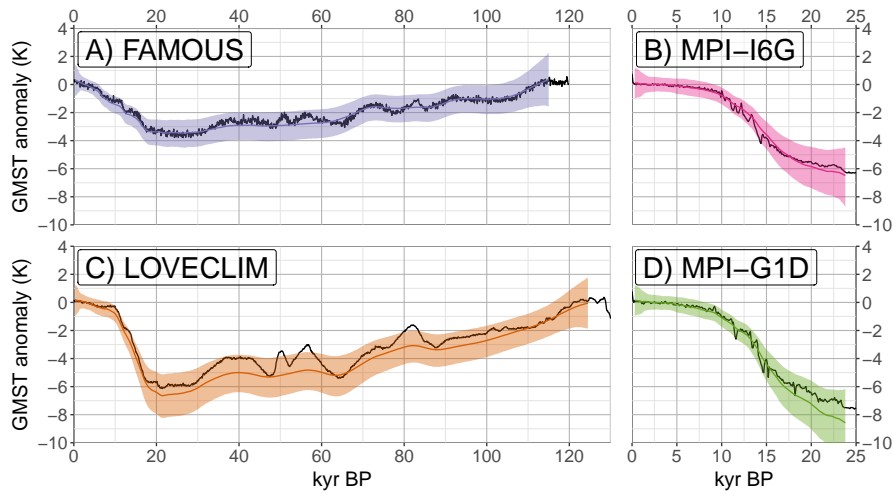

**Figure 3.** Reconstructed GMST based on the pseudo-proxies from the *Full PP* experiment. The coloured lines are the mean of the reconstructed GMST ensemble, while the shading corresponds to the 90% confidence interval. We consider here the mean across the ten computed members. The black lines are the simulated GMSTs. The anomaly is computed with respect to the late Holocene (0 to 5 kyr BP).

## 4 Results

The various pseudo-proxy experiments are analysed in this section. We start with the *Full PP* experiment, whose results are presented in Fig. 3. This experiment gives a general overview of the ability of the GMST reconstruction algorithm. In particular, the algorithm successfully reconstructs the orbital timescale variations of the four simulations. The mean pseudo-proxy reconstruction presents however a strong smoothing of sub-orbital timescales. Finally, the temperature anomaly of the glacial period in the pseudo-proxy reconstructions is, for most simulations, slightly too cold, although the targeted simulated GMST is within the reconstruction uncertainty range. In light of these preliminary results, we arrange the analysis of the results around three main topics, addressed in the following subsections: quantification and origin of the uncertainty, amplitude and timing of orbital timescale variations, and representation of the timescale continuum.

### 4.1 Quantification and origin of the uncertainty

#### 4.1.1 Uncertainty range as estimated by the algorithm

The GMST reconstruction algorithm estimates uncertainties using a Monte Carlo approach. Five different sources of uncertainty are estimated at different stages of the algorithm: measurement and calibration noise, age model uncertainty, selection of records (which we interpret as a location resampling), latitudinal band configurations, and scaling factor. The uncertainty is propagated to the GMST reconstruction by considering an ensemble of realisations.





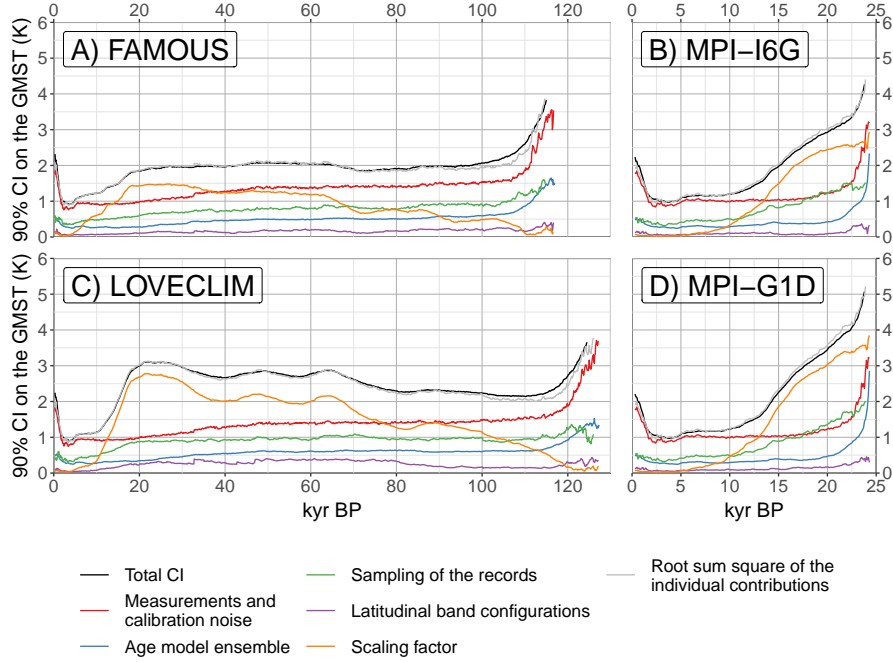

**Figure 4.** Estimation of various sources of uncertainty for the *Full PP* experiment. The uncertainty is quantified as the width of the 90% confidence interval (CI) of the reconstructed GMST ensemble. The black lines correspond to the CI in fig 3. The coloured lines are the individual contributions of each source of uncertainty quantified in the GMST reconstruction algorithm. Note that we interpret the CI arising from the selection of record as a location resampling. The grey line is the root sum square of the individual contribution. It should correspond to the black line if the individual contributions are independent of each other.

The width of the total 90% confidence interval (CI) is presented in Fig. 4, for the various simulations, alongside an estimation of the individual contributions of the considered uncertainty sources. We first notice a large uncertainty increase at both ends of the time series. The Holocene has otherwise the lowest uncertainty ($\sim 1\,\mathrm{K}$ for all simulations). The uncertainty is maximum during the LGM ($2-4\,\mathrm{K}$). We also find that the root sum square of the individual contributions is very close to the total CI.

This implies that reducing the uncertainty from the highest sources of uncertainty is the most impactful.

One of the most important contributors to the uncertainty estimate is the scaling factor. By definition, it is proportional to the temperature anomaly, and exhibits, therefore, large fluctuations. It is negligible during the Holocene and the LIG, but accounts from 62% (FAMOUS) to 80% (LOVECLIM) of the uncertainty estimate during the LGM, depending on how cold the simulated LGM is. On average, the most important contributor to the uncertainty estimate is however the measurement

and calibration noise, with a CI width mostly between 1 and $1.5\,\mathrm{K}$, and an average contribution from 38% (LOVECLIM) to 50% (FAMOUS, MPI-I6G). It also increases sharply at both ends of the time series and is the main contributor to the same behaviour in the total CI. The CI width of the location resampling and the age model are below 1 and around $0.5\,\mathrm{K}$ respectively





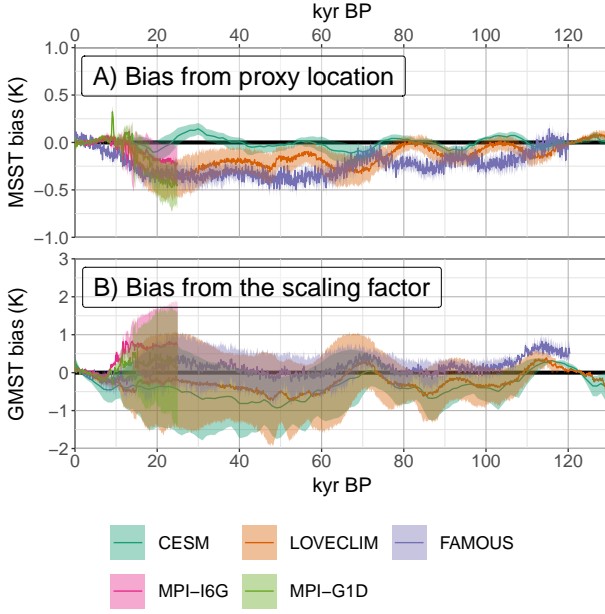

**Figure 5.** Potential reconstruction biases and evaluation of the confidence interval. In panel A, the $60°$S-$60°$N mean SST (MSST) recon-struction from the experiment *SST at proxy locations* is compared to the simulated MSST. The shaded area is the 90% CI, only estimating uncertainty from location resampling and the latitudinal band configuration. A properly quantified confidence interval should overlay the 0 line 90% of the time. In panel B, we apply the scaling factor with uncertainty to the simulated MSST. We then compare the result to the GMST. As above, a properly quantified confidence interval should overlay the 0 line 90% of the time.

(or, in terms of average contribution, around 15% and 6% for all simulations). Finally, the configuration of the latitudinal bands has the smallest impact on the uncertainty estimate (below $0.5$ K, or 1%).

### 4.1.2 Ability of the algorithm to estimate the uncertainty

We further the analysis by investigating whether the GMST reconstruction algorithm correctly estimates the uncertainty sources, starting with the measurement and calibration noise. By definition, the 90% CI should have a coverage frequency of the targeted signal of 90%. However, using the *measurement noise* experiment, we find that the 90% coverage frequency is already reached by a CI, smaller than the 90% CI by a factor of 2 (FAMOUS) to 2.3 (All the others). The overestimation of the CI is not unexpected: the measurement noise introduced in the pseudo-proxies by *sedproxy* is between 0.23 and $1$ K (cf. tab. 1), while the noise introduced in the reconstruction algorithm is $1.5$ K. This overestimation is a conservative approach in case the measurement and calibration uncertainty is higher in reality than what is used to compute the pseudo-proxies.

Next, we evaluate the ability of the reconstruction algorithm to estimate the $60°$S-$60°$N mean SST (MSST, before applying the scaling factor) based on the specific set of proxy locations made available in the J20 dataset. To that end, we compare the MSST reconstructed from the *SST at proxy locations* experiment to the simulated MSST (Fig. 5-A). We find that the





reconstruction for all simulations but CESM exhibits a cold bias during the LGM (up to $0.5\,\mathrm{K}$) and most of the glacial period. In addition, location resampling and latitude band configurations, which aim to account for it, are not large enough to cover the bias. Yet, the pseudo-proxy experiments using random proxy locations can reproduce the simulated MSST. Therefore, the bias is caused by the specific set of locations in the J20 dataset: there is an over-representation of regions with strong LGM cooling (e.g. North-West Atlantic and origin of the Kuroshio extension current), compared to regions with weaker to no cooling (e.g. North polar gyre and eastern part of the subtropical gyres in the Pacific) in the same latitudinal band (cf. Fig. 1-A). In addition, how the uncertainty from the location sampling of records is estimated relies on the hypothesis that the temperature anomalies are well-distributed, which is not the case, explaining the underestimation of the uncertainty range.

The last uncertainty estimate we evaluate is the scaling factor (Fig. 5-B). We find the errors constrained within $1\,\mathrm{K}$. The coverage frequency of the 90% CI ranges between 46% (MPI-I6G) and 79% (LOVECLIM), which denotes a small underestimation. By definition, the uncertainty of the scaling factor is proportional to the temperature anomaly. We suggest that this overly strong constraint leads to an overestimation (underestimation) of the uncertainty for large (small) temperature anomalies. This can be tackled by considering in addition an additive error independent of the SST anomaly.

### 4.1.3   Factors influencing the uncertainty estimates

We continue the analysis by investigating the dependency of the uncertainty estimates. Using the experiments *Full PP at random locations*, we find the number of records considered to strongly impact the uncertainty. Using 61 proxies as a baseline (number of records in S16), we find that 83 records (average number of records between MIS3 to 5) reduce the uncertainty range of the reconstruction by ∼15%. With 200 (as during the deglaciation) the reduction reaches 46%. In case 400 records were available, the reduction would be 62%. This reduction mostly affect the band configurations and the measurement noise, followed by the location resampling and age uncertainty (in relative values). The proxy number has of course no effect on the scaling factor.

The variation of the number of records through time in the *full PP* experiment (Fig. 2-A) explains most of the variations in the uncertainties shown in figure 4, and in particular the increase at both ends of the time series Note that the number of records considered in the PPE at both ends of the timeseries is not only related to the number of records available in J20 (cf. Fig. 2-A), but also to the fact that some pseudo-proxy values cannot be computed as age uncertainty spans beyond the timespan of the simulations. This is most evident in the MPI simulations (Fig. 4-B and D). In addition, measurement noise also depends on the age resolution, as a higher number of measurements reduces the noise introduced in the reconstruction algorithm. The location sampling uncertainty depends on the homogeneity of the temperature anomaly field, that is, to the first order, on the GMST anomaly. Finally, uncertainty due to the age model is dependent on the rate of change of the GMST: abrupt changes such as the deglaciation, or the millennial-scale variability exhibited in the MPI simulation or LOVECLIM increases the uncertainty. However, these effects only become noticeable once the number of records is fixed.





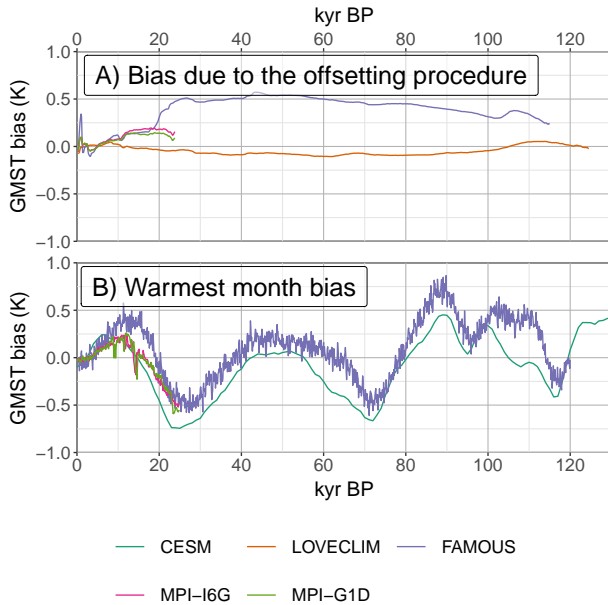

**Figure 6.** Other potential sources of biases, non-quantified in the uncertainty range. In panel A, the GMST reconstruction from the *Full PP* experiment is compared to the one of the *no offsetting* experiment. Positive values mean that the offsetting procedure produces a too warm reconstruction. In panel B, the GMST reconstruction from the *warmest month* experiment is compared to the one of the *SST at proxy locations* experiment. Positive values mean that reconstructions based on proxies biased towards the warm season are too warm.

### 4.1.4 Potential sources of uncertainty non-quantified by the algorithm

Finally, we investigate other sources of uncertainty, which are not or only partly estimated by the reconstruction algorithm. We quantify, in particular, the effect of the offsetting procedure and potential seasonal bias (Fig. 5). The offsetting procedure
generates errors generally below $0.2\,\mathrm{K}$ on the reconstructed GMST, except for FAMOUS, where it reaches $0.5\,\mathrm{K}$. We suggest this bias is reasonable given the number of records it enables the algorithm to additionally consider (96).

We also estimate an upper bound to the effect of seasonality bias on the pseudo-proxy reconstruction. If all records were not recording the mean annual temperature, but that of the warmest month, it would generate biases up to $0.75\,\mathrm{K}$. This bias mostly follows the northern hemisphere insolation curve: warm during the early Holocene, cold during the LGM and MIS4,
and warm again during MIS5. These orbital-scale biases can significantly impact the evaluation of orbital timescale variations, in particular regarding the LGM cooling which it could overestimate, or an early Holocene warm period which would only appear for the summer months.

In conclusion, we reckon that the estimation of the uncertainty range by the algorithm is realistic, but that the estimations of the individual contributions are biased and should be used with caution. We suggest that the measurement and calibration
noise, the set of record locations, and the scaling factor contribute overall similarly to the uncertainty.





**Table 3.** Quantitative assessment of the reconstruction of the amplitude and timing of orbital timescale variations. We use the GMST reconstructed for the *Full PP* experiment. The LGM temperature anomaly is computed for LOVECLIM and FAMOUS as the coldest 4 kyr later than 40 kyr BP with respect to the last 5 kyr. The same definition is used for LGM occurrence. Due to the difficulty of defining the coldest period in the MPI simulations, we use the average between 19 and 23 kyr BP instead. The Deglaciation end is defined as the last time the temperature anomaly reaches one-tenth of the LGM temperature anomaly. Similarly, glacial onset is defined as the first time the temperature anomaly falls below one-tenth of the LGM temperature anomaly. Finally, we also compute a temperature anomaly for the end of the LIG, as the average between 119 and 123 kyr for LOVECLIM and between 110 and 114 kyr BP for FAMOUS. "Mean" is the mean value across the reconstruction ensembles. "Robustness" is the range of the mean value for each of the ten ensembles computed for the experiment. It characterises the robustness of the mean reconstruction to random factors in the proxy data. "90% CI" is the mean 90% confidence interval, as calculated by the GMST algorithm, within each reconstruction ensemble. The target is the actual simulated value.

|  | LOVECLIM | | | | FAMOUS | | | |
|---|---|---|---|---|---|---|---|---|
|  | mean | robustness | 90% CI | target | mean | robustness | 90% CI | target |
| End of LIG anomaly (K) | -0.4 | -0.68/-0.05 | -1.32/0.56 | -0.09 | -0.04 | -0.18/0.17 | -1.03/0.93 | -0.03 |
| Glacial onset (kyr BP) | 118.2 | 116.8/119.7 | 113.5/122.9 | 118.3 | 106 | 102.5/108.4 | 93.1/113.2 | 109.9 |
| LGM anomaly (K) | -6.65 | -6.89/-6.39 | -7.95/-5.40 | -5.97 | -3.6 | -3.75/-3.43 | -4.39/-2.85 | -3.57 |
| LGM occurrence (kyr BP) | 23.3 | 22.6/24 | 20.9/27 | 22.4 | 25.7 | 24.4/26.8 | 19.8/32.2 | 25.3 |
| Deglaciation end (kyr BP) | 8.7 | 8.1/9.2 | 6.5/10 | 9.9 | 5 | 4.7/5.2 | 4/6.2 | 5.4 |
|  | MPI-I6G | | | | MPI-G1D | | | |
| LGM anomaly (K) | -6.02 | -6.13/-5.83 | -7.24/-4.85 | -5.69 | -7.67 | -7.82/-7.46 | -9.17/-6.25 | -6.72 |
| Deglaciation end (kyr BP) | 8.9 | 8.6/9.3 | 7.1/10.2 | 9.2 | 8.8 | 8.5/9.2 | 7.4/9.9 | 9.2 |

## 4.2 Amplitude and timing of orbital timescale variations

Given the J20 dataset, one of the key characteristics we expect the algorithm to reconstruct accurately is the amplitude and timing of orbital timescale variations of the simulated GMST. While these characteristics can be qualitatively assessed in Fig. 3, we also provide quantitative metrics with uncertainty and robustness (Tab. 3). Note that we use slightly different definitions
for the LGM and the end of the LIG to accommodate the period and behaviour of the simulations (cf. caption).

As already discussed, a cold bias is evident for LGM and end of LIG temperature in all simulations but FAMOUS, but the true anomalies remain included in the 90% CIs. This bias is mostly caused by the location bias (cf. Fig. 5-A), while the other factors (offsetting procedure, scaling to GMST and seasonality) either reduce or amplify it.

We also investigate the timing of various events: Glacial onset, LGM, and deglaciation end. These metrics are only designed
to compare the result of the pseudo-proxy reconstruction to the simulations. We find the reconstructions to match very closely the timing of the simulations. We further suggest that the 90% CI as calculated by the GMST algorithm largely overestimates the uncertainty. While the overestimated measurement noise helps to account for non-quantified sources of temperature bias in the GMST algorithm, it may also lead to an overestimation of the timing uncertainty, where no biases are evident. Interestingly,



the PPE including only bioturbation shows a lag of about half the bioturbated layer, which is not evident in the table. This bias
is caused by the non-symmetric shape of the function characterising the sediment surface mixing.

Finally, we find our result robust to different realisations of pseudo-proxies. The measure of robustness also describes a
limit under which temperature and timing differences depend on the pseudo-proxy realisation: about $0.5\,\mathrm{K}$ and from $0.7\,\mathrm{kyr}$
in the Holocene to $5\,\mathrm{kyr}$ during the LIG. Interestingly, these values are below the uncertainty associated with individual proxy
measurements. In addition, we suggest that this measure characterises the uncertainty better than the CI provided by the GMST
algorithm, when comparing values to one another.

### 4.3 Representation of the timescale continuum

As noted in Fig. 3, the algorithm does not reconstruct the high-frequency variability of the simulated GMST. We further
investigate this loss of signal using spectral densities and coherence. We focus on the MSST as the scaling factor does not
influence the high-frequency variability. Figure 7 compares the spectrum of variability of the simulated MSST to the one
reconstructed in several PPEs. We first focus on the *Full PP* experiment. The individual Monte Carlo realisations of the
reconstruction (light grey lines) appear to overestimate the power spectral density (PSD) for most timescales but the longest
ones. In particular, their PSD exhibits a straight slope for the shorter timescales ($\sim -2\,\mathrm{K}^2$). The overestimated spectrum is
mostly related to the large measurement noise introduced in the GMST algorithm. By contrast, the spectrum of the ensemble
mean (darker grey lines) underestimates variability at most timescale, with a steeper slope (between $-2.7\,\mathrm{K}^2$ for FAMOUS
and $-3.3\,\mathrm{K}^2$ for the MPI simulations). The PSD is properly represented only for timescale longer than $10\,\mathrm{kyr}$ for the entire
LGC, and longer than $4\,\mathrm{kyr}$ for the last $25\,\mathrm{kyr}$ (the latter holds regardless of the simulation). None of the pseudo-proxy
reconstructions are able to represent all the characteristics of the simulated PSD, such as the high variability at short timescale
in FAMOUS, or the high variability at millennial timescale in the MPI simulations. The loss of PSD between the ensemble
mean and each member's spectrum is due to a lack of coherence between reconstruction members (i.e. loss of signal).
We further investigate the cause of the underestimated variability of the ensemble mean with sensitivity PPEs. We iden-
tify four potential causes: the bioturbation, which smooths the local SST time series, the age uncertainty, which reduces the
coherence between the local SST time series, the age resolution, which permits resolving the shortest timescale, and the mea-
surement and calibration noise in *sedproxy*, which adds non-climatic variability (cf. coloured lines in Fig. 7). The PSD from
the *measurement noise* experiment (yellow) is the closest to the simulated one with just slight overestimation, but lose the
simulated PSD characteristics below $1\,\mathrm{kyr}$. In this PPE, the noise is injected at the centennial scale, resulting in increased
variability at this timescale and upward propagation. Over the LGC, none of the other PPEs is able to represent the simulated
spectrum below $10\,\mathrm{kyr}$, with the strongest loss of PSD due to age uncertainty (blue). For the last $25\,\mathrm{kyr}$, the PPEs are still
able to reproduce the large multi-millennial variability, which is only lost by the joint consideration of bioturbation and age
uncertainty (purple).
The loss of signal at specific timescales can be further characterised using coherence. Coherence, or the squared coherency
spectrum, is a statistical method to analyse the dependency between two time series through the timescales (Von Storch and
Zwiers, 1999). It can be seen as a timescale-dependent squared correlation. A coherence of 0 implies that the two time series



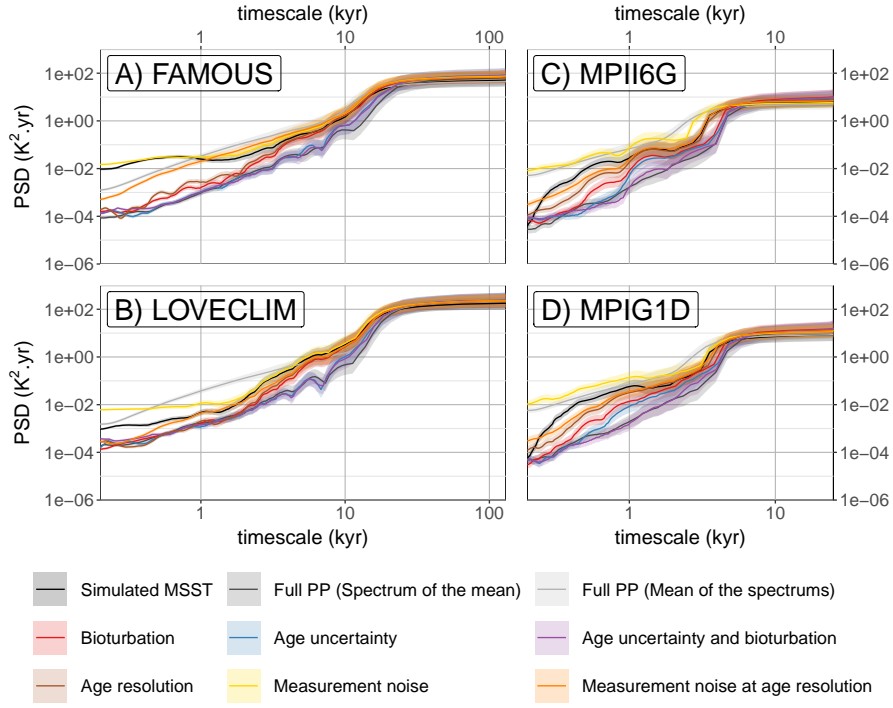

**Figure 7.** Loss or gain of variability at specific timescales between reconstructed and simulated MSST. The black line is the Power Spectral Density (PSD) of the simulated MSST. The other lines correspond to various PPEs defined in tab. 2. For the *Full PP* experiment, we consider both the PSD of the ensemble mean, and the PSD of each ensemble member. Only the ensemble mean is considered for the other PPEs. Finally, the shading corresponds to the confidence interval at the level 0.05 of the PSD estimates.

vary independently at this timescale. A coherence of 1 means that the time series vary synchronously although a lag can exist. The coherence also does not characterise the amplitude of the variation, which we have already investigated with the spectral

densities. Fig. 8 presents the coherence between the simulation and reconstructed MSST from the same PPEs as in Fig. 7. For the *Full PP* experiment, coherence is very high (> 0.9) for the longest timescales in all simulations. The coherence decreases however sharply for timescales shorter than $15 \, \text{kyr}$ for the LGC simulations (FAMOUS and LOVECLIM), which corresponds to the start of the PSD loss in Fig. 7-A and C. The drop happens at shorter timescales (below $4 \, \text{kyr}$) for the last $25 \, \text{kyr}$, again corresponding to the timescale on which the PSD loss starts. A similar timescale has been found for the Holocene (Dolman

et al., 2021).

We further the analysis with the sensitivity PPEs. We find the age uncertainty (blue) to be the main cause of the loss of coherence on the LGC. For the last $25 \, \text{kyr}$, bioturbation also plays an important role, but neither can individually explain the coherence loss exhibited by the *Full PP* experiment. Measurement noise (yellow) and age resolution (brown) have the smallest impact on the coherence for either the LGC or the last $25 \, \text{kyr}$. However, the measurement noise is in reality linked to the age

resolution. The PPE considering both (orange line) exhibits a coherence loss at a timescale close to the age uncertainty and





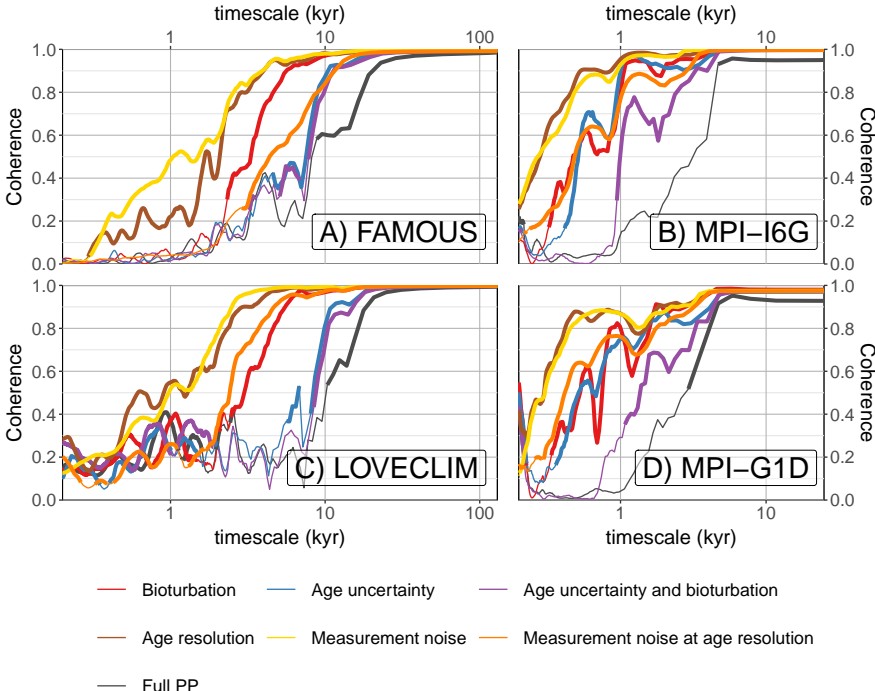

**Figure 8.** Disentangling the origin of the loss of signal at high frequencies. The loss of signal is quantified by the coherence between the simulated and reconstructed MSST for various PPEs. For the latter, we use the ensemble mean. The thicker lines correspond to significant values at the level of 0.05. Specifically, they are values above the 95 percentile of the coherences between the simulated MSST and randomly phase-shifted time series of the same characteristics.

bioturbation, suggesting it also has a key role for the last $25\,\mathrm{kyr}$. Note that, for the number of records considered, the set of record locations does not impact the coherence significantly. Simulated GMST and MSST are also significantly coherent at all timescales (not shown).

Our results show that the difference in results between the LGC and the last $25\,\mathrm{kyr}$ is directly related to the dataset char-
acteristics (Fig. 2-B to D). In the last $25\,\mathrm{kyr}$, the average 90% CI of the age uncertainty and the average bioturbated layer have similar values, corresponding to the timescale when coherence is lost. Age uncertainty, resolution and bioturbated layer increase beyond this time period, but the increase in age uncertainty is the most important. This one reaches $\sim 10\,\mathrm{kyr}$ beyond $40\,\mathrm{kyr}$ BP, again similar to the timescale when coherence drops.




## 5   Discussion

The thorough investigations of the previous section allow us to discuss the performance of the reconstruction algorithm applied to the J20 dataset (Sect. 5.1). Nevertheless, future avenues for these pseudo-proxy analyses are possible (Sect. 5.2). We also discuss which limiting factors from either the algorithm (Sect. 5.3) or the dataset (Sect. 5.4) should be addressed in priority to improve the GMST reconstruction quality.

### 5.1   Quality of the reconstruction

According to our analysis of the PPEs, the timing of GMST variations can be reconstructed for timescales longer than $4\,\mathrm{kyr}$ and a timing uncertainty of $\pm 0.5$ to $1\,\mathrm{kyr}$ for the last $25\,\mathrm{kyr}$. Beyond $40\,\mathrm{kyr}$ BP, only timescale longer than $15\,\mathrm{kyr}$ can be reconstructed, with an uncertainty of up to $\pm 3\,\mathrm{kyr}$. This precision of the reconstruction, and its variation through time, is directly related to the underlying dataset characteristics. The reconstruction of shorter timescales beyond $40\,\mathrm{kyr}$ BP is limited by the age uncertainty of the records. For the last $25\,\mathrm{kyr}$, the limitation originates from a combination of age uncertainty,
bioturbation and measurement and calibration noise.

Most reconstructed pseudo-proxy GMSTs also exhibit a cold bias of up to $1\,\mathrm{K}$ during the glacial period, which we attribute to the specific set of record locations in J20. Uncertainty on the temperature anomaly is otherwise strongly dependent on the number of proxy records and the scaling factor. The GMST algorithm estimates the uncertainty range (90% CI) from less than $1\,\mathrm{K}$ during the Holocene, to up to $3\,\mathrm{K}$ for a cooling of $6\,\mathrm{K}$ during the LGM. Our results suggest, however, that the
Holocene uncertainty is overestimated because of an overestimated measurement noise uncertainty. Similarly, the uncertainty for large temperature anomalies is also likely overestimated due to the definition of the scaling factor uncertainty. For the LIG, overestimated measurement noise and underestimated scaling uncertainty seem to cancel each other and lead to reasonable results. These overestimations help nevertheless to account for uncertainty sources not properly accounted for, such as locations or potential seasonality bias. In particular, an underestimation of the seasonality bias could lead to a deterioration of the
performance, both in terms of amplitude and timing of variations, especially for the period between the LGM and the early Holocene, where uncertainty is otherwise the lowest.

The analysis of the uncertainty of the timing and amplitude of variations shows that there exists a trade-off between including a large number of records, which helps to reduce the biases and uncertainty on the amplitude, and including only the records of the highest quality (low age uncertainty, high accumulation rate, high resolution), which help better resolve the shorter
timescale variability. While a finer selection of records could balance the two, we also suggest that the algorithm could handle records of different characteristics better (cf. Section 5.3).

These results are robust across the set of simulations and we assume them to hold for the real-world reconstruction. In addition, while these results are specific to the adapted S16 algorithm applied to the J20, we expect that the general concepts discussed here hold for any aggregation algorithm and dataset. In particular, the algorithm could be used for regional recon-
structions (as done in Clark et al., 2024), as long as a sufficiently high number of records are considered. We suggest, for



example, that the accurate reconstruction of the timing of orbital timescale variations ($> 10\,\text{kyr}$) offers a good opportunity to investigate the synchronicity of regional temperature anomalies and forcings.

## 5.2 Advances in pseudo-proxy experiments

Our results rely on advances in pseudo-proxy experiments, which were enabled by the availability of proxy system models and climate simulation output.

### 5.2.1 Model data

In addition to the proper availability of simulated climate fields for the production of PEE, the PPE results can only be transferred to real proxy data if these fields are realistic enough. The recent advances in computing capacities, modelling, and our understanding of climate drivers have enabled the production of more realistic climate simulations over longer periods of time (Ivanovic et al., 2016). Yet, the PPEs we conduct require data only available for a handful of simulations. However, a multi-model framework for PPEs is key to assessing our results' robustness.

One of the requirements is of course the simulated time period, which needs to cover more than the period of interest, in order to take into account the age uncertainty of the records. For example, properly investigating the LIG requires data from the penultimate deglaciation. Another requirement is that these simulations must include transient forcings, so that realistic variations of the GMST can be produced at sufficiently high resolution. In particular, centennial or lower resolution is needed here to resolve the effect of bioturbation. The addition of other forcings such as meltwater in the MPI simulations also increases the degree of realism of the simulated GMST (Weitzel et al., 2024). These additions make more refined analysis of the results possible, which can more easily be transferred to real proxy data. For instance, we can assess that the GMST algorithm is not able to reconstruct the effect of melt-water forcing, such as the abrupt events during the deglaciation.

Finally, the spatio-temporal variability of the simulated climate field affects our results. For example, the amplitude of the temperature bias due to the proxy locations directly depends on the spatial pattern of the simulated temperature field. Differences between the simulations lead to differences in the estimation of the bias in the PPE reconstruction (cf. Fig. 5-A). Therefore, despite some agreements between simulations on the sign and amplitude of the bias, proper quantification of this bias for the real-world reconstruction would first require an evaluation of the degree of realism of the spatio-temporal patterns. In support of this, the temperature field of simulations is known to be more homogeneous at the regional scale compared to proxy records (Jonkers et al., 2023). This has been demonstrated for the multidecadal to multimillennial-scale (Laepple et al., 2023; Weitzel et al., 2024). For the orbital timescale ($> 10\,\text{kyr}$), Paul et al. (2021) suggested that upwelling regions, where many records come from, can have a different variability while being too small to be properly resolved by the models considered here. The use of too homogeneous fields can lead to an underestimation of location biases on the reconstructed temperature and overconfidence in the algorithm.





### 5.2.2 Proxy system models

In PPEs, proxy system models are used to produce as realistic as possible pseudo-proxy values. However, proxy system models only represent our current understanding of the proxy system, which limits the extrapolation of our results to the real-world reconstruction. Yet, the use of *sedproxy* presents the advantage, compared to more conceptual approaches (e.g. Wang et al., 2014; Jaume-Santero et al., 2020; Nilsen et al., 2021; Weitzel et al., 2024), to model specific processes, such as the measurement and calibration noise and bioturbation, as a function of the record's metadata. Sensitivity PPEs can therefore be computed to evaluate the impact of these processes.

Yet, our results are also limited by the proxy system model used. For example, the impact of the processes occurring during the *sensor stage* for $TEX_{86}$, LDI or assemblages has only been crudely accounted for within a generic measurement error. We therefore did not investigate in depth the influence of the proxy types on the reconstruction. While increased uncertainty from the measurement or the calibration will reduce the ability of the reconstruction to resolve the shorter timescale, other biases affecting the longer timescale would require a more thorough analysis. These potential biases include for example the effect of any variables other than the SST, such as depth, salinity or seasonality, on the temperature signal recorded (Telford et al., 2013; Timmermann et al., 2014; Ho and Laepple, 2015; Jonkers and Kučera, 2017). The seasonality bias is considered for most species recording $Mg/Ca$ ratios in *sedproxy*, but assumed negligible for the other proxy types. For this reason, we design a PPE to estimate an upper bound of the effect of seasonality, where all proxies record the warmest month of the year. This warm season bias has, for example, been one of the main hypothesised reasons for the model-data discrepancies during the early Holocene (Liu et al., 2014; Marsicek et al., 2018; Bova et al., 2021), although there is also evidence for cold season bias for other periods and species (Steinke et al., 2008; Timmermann et al., 2014).

### 5.3 Improvement of the algorithm

One of the primary objectives of this paper is to provide a framework to evaluate the performance of GMST reconstruction algorithms. We focus on an adaptation of S16 as an example. The adaptations are minor, and only required to improve the use of the J20 dataset and the characterisation of the algorithm's performance (cf. sect. 3.1.2). Yet, our evaluation makes evident avenues of improvement for the algorithm which we discuss here.

We have already discussed how the algorithm is sensitive to the spatial distribution of proxy records. This sensitivity relates to the assumption that temperature anomalies are similar within a latitudinal band. Both the simulations used here and other proxy analyses show that this is not the case (e.g. Judd et al., 2020; Tierney et al., 2020; Paul et al., 2021). Some studies have used more complex methods and relied on either present SST observations (e.g. Paul et al., 2021), or climate simulations (e.g. Osman et al., 2021; Annan et al., 2022). These methods rely however on the assumption that the spatial covariance of present-day does not change through time, or that of the simulated SST is similar to the reality. The design of another aggregating method could help to reduce the influence of areas with a large density of records, without relying on external datasets.

Our refined algorithm relies on the stacking of proxy records to increase the signal-to-noise ratio. However, records are stacked together, regardless of their quality (age uncertainty, accumulation rate, resolution). This stacking method leads to a



trade-off between data quality which improves the reconstruction of the shorter timescales, and a high number of records, which reduces the overall uncertainty (cf. 5.1). A new stacking method could be researched to limit the impact of this trade-off by, for example, taking into account the timescale resolved by each record.

Finally, the scaling of the MSST to the GMST was introduced by S16 due to the limited availability of reliable local temperature reconstruction over land and ice-covered areas for the investigated time period. This scaling is however a critical source of uncertainty, especially concerning the amount of glacial cooling or LIG temperature anomaly. This has direct consequences on our ability to constrain Earth system sensitivity or the global response to climate forcings. The transient simulations used here show that the variation of GMST and MSST is not completely linear. This behaviour suggests going beyond the simple LGM-to-PI ratio used to compute the scaling factor in S16. Despite their scarcity, the use of terrestrial proxies for temperature could still help to better characterise the scaling. Yet, large areas are devoid of proxy information, in particular over sea ice and ice sheets that have melted. Physics-based constraints remain needed for these areas.

## 5.4 Limiting factors from the dataset and future developments

Expanding the dataset with new records, in particular for the periods and areas where fewer records are available, will evidently improve the reconstruction. Nevertheless, spatial-temporal inhomogeneity will always remain due to geological constraints. Other avenues can improve the quality of and the confidence in the reconstruction, from the perspective of the proxy dataset. First, the age uncertainty is the limiting factor preventing the reconstruction of multi-millennial timescale beyond $30\,\mathrm{kyr}$ BP, and the main focus should be put on reducing this uncertainty (e.g. Waelbroeck et al., 2019; Peeters et al., 2023). In addition to improving the confidence in the absolute age, information on the relative age between records can also be provided, if properly accounted for by the reconstruction algorithm.

Second, measurement and calibration noise and bioturbation are another source of loss of signal on short timescales, particularly in the last $30\,\mathrm{kyr}$. Here, we suggest better quantifying these uncertainties for each record, so that the algorithm can sort the records depending on the timescale they can resolve. In particular, bioturbation depth could be quantified for each record, and measurement uncertainty could depend on the number of replications or sample size for each data point.

Finally, we find that potential seasonality bias can significantly decrease the accuracy of the reconstruction. For many of the records available, it is unclear whether the reconstructed temperature suffers for a seasonal bias. Corrections from seasonally biased records have been suggested (e.g. Bova et al., 2021), but rely on assumptions that must be verified beforehand (Laepple et al., 2022).



## 6   Conclusions

In this study, we design a framework for the evaluation of algorithms reconstructing spatial mean temperatures. Our framework relies on recent advances in pseudo-proxy experiments, which include the availability of realistic proxy system models and long transient climate simulations. We apply the framework to an adapted version of the GMST reconstruction algorithm used in S16, and the synthesis of marine proxy records for temperature of the LGC from J20. The quantitative results presented here can serve as a basis for future evaluations of LGC reconstructions, while the framework can be applied to other aggregation-based reconstruction algorithms and other datasets.

Our results are based on PPEs computed from an ensemble of 4 transient simulations of the LGC or of the last $25\,\mathrm{kyr}$. We find that the pseudo-proxy reconstructions based on the S16 algorithm and the J20 dataset perform differently over time. For the last $25\,\mathrm{kyr}$, the temperature variations and their timing are accurately reconstructed for timescales of at least $4\,\mathrm{kyr}$. Sensitivity PPEs show that age uncertainty, bioturbation and measurement noise smooth the reconstruction, leading to a loss of signal below this timescale. Uncertainties remain large on the amplitude of temperature variations, even for longer timescales, although we find the algorithm to overestimate it. The reconstructions exhibit in particular a cold bias for most model simulations, which is related to the non-uniform distribution of record locations. Other sources of uncertainty include the scaling of the mean SST to the GMST, the measurement and calibration noise, and potential seasonality bias. The number of records plays a critical role in reducing all uncertainty sources but the scaling. The decreasing proxy number and increasing uncertainty on the scaling are the two main factors explaining the increased uncertainty from the Holocene to the LGM. Beyond $40\,\mathrm{kyr}$ BP, only timescales longer than $15\,\mathrm{kyr}$ can be reconstructed, due to a sharp rise in age uncertainty. We assume these results to hold for real-world reconstructions of the LGC.

Our results show also the existence of a trade-off between the inclusion of a large number of records, which overall reduces the uncertainty, and of only the highest quality records (low age uncertainty, high accumulation rate, high resolution), which improves the reconstruction of the short timescale. The reconstruction could be improved by a better filtering of the input record data, or by a better handling of the varying record quality by the algorithm. We also suggest other avenues of improvement for the algorithm, to handle better the spatial aggregation and the scaling to GMST. From the proxy record perspective, reducing the age uncertainty is the most critical challenge to tackle.



*Code and data availability.* The R code and the proxy metadata (subset of hte PalMod database) to reproduce the results and plots of this study are available at https://doi.org/10.5281/zenodo.11126855. All simulation datasets are also available online: https://www.wdc-climate.de/ui/entry?acronym=PMMXMCRTDIP122 (MPII6G), https://www.wdc-climate.de/ui/entry?acronym=PMMXMCRTDGP122 (MPIG1D), https://data.ceda.ac.uk/badc/quest/data/quaternaryq/famous_glacial_cycle (FAMOUS), http://climatedata.ibs.re.kr:9090/dods/public-data/loveclim-784k
(LOVECLIM), http://climatedata.ibs.re.kr:9090/dods/public-data/pcesm-3ma/pcesm-3ma-jan (CESM).

*Author contributions.* KR, NW and JPB designed the study. JPB processed the data and implemented the pseudo-proxy experiments with assistance from MM and input from AMD, LJ and NW. All authors discussed the results. JPB wrote the manuscript. All authors commented on earlier drafts of the manuscript and approved its final version.

*Competing interests.* The authors declare that they have no conflict of interest.

*Acknowledgements.* We thank Marie-Luise Kapsch and Uwe Mikolajewicz for their help in providing the data of the MPI simulations before their publication, as well as their insightful comments and support through the development of the study. We also thank the teams responsible for the production of the FAMOUS, CESM and Loveclim simulations, and to have provided the output online.

This research was funded by the German Federal Ministry of Education and Research (BMBF) within the Research for Sustainability initiative through the PalMod project (grant no. 01LP1509C, 01LP1922A, 01LP1926C, 01LP2308A, and 01LP2311C), and by the Deutsche
Forschungsgemeinschaft (DFG, German Research Foundation; project no. 395588486)

AMD was also supported by the Deutsche Forschungsgemeinschaft (DFG, German Research Foundation; Project no. 468685498; SPP 2299/Project no. 441832482). KR is also a member of the Machine Learning Cluster of Excellence, funded by the Deutsche Forschungsgemeinschaft (DFG, German Research Foundation) under Germany's Excellence Strategy (EXC number 2064/1; Project no. 390727645).



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
