# Peer review of "Testing the reliability of global surface temperature reconstructions of the last glacial cycle with pseudo-proxy experiments"

_EGUsphere, 2024_

## Referee Comment (RC1)

**Review of Baudouin et al (2024)**

For questions regarding this review, please contact bryan.lougheed@geo.uu.se (until 31/08/2024, thereafter @outlook.com).

**Summary of review**

This paper further investigates an existing algorithm for global mean surface temperature (GMST) that was developed by Snyder (2016) using existing palaeoclimate data. Here, the authors compare the data-based GMST reconstruction by Snyder to a model-based reconstruction of GMST. Specifically, the authors have taken climate model output and passed it through proxy and sediment models, thus producing a type of synthetic palaeoclimate data, which can then be compared to the Snyder analysis of palaeoclimate data.

In general I agree that processes in the sediment archive itself (such as bioturbation) were underestimated by Snyder, so an approach using synthetic palaeoclimate data is useful. In general I find the results to be interesting and the sty to be a useful addition to the literature, but I think the paper could due with some re-writing so that it states what is being done in a more straightforward and accessible way. This would especially benefit readers who are less well familiar with the subject matter. I found the reading quite heavy in places and not so logically organised. There are also very many acronyms.

I also think that the study can be better framed in the abstract and introduction. In particular much of the paper is framed from a point of view of evaluating the Snyder et al GMST method by comparing it to the authors' synthetic palaeoclimate data, thus assuming the authors' data represents the truth. But in reality both methods represent an approximation, and we don't know which is the "truth". I agree that the method of the authors attempts to include more processes, but a more accurate framing of the paper would be to compare the two methods, rather than to use one method to evaluate the other.

Also, throughout the paper I found that the citations skew to the very recent past, with many older papers being forgotten.

**Specific comments (authors' text is in blue)**

Lines 1-3: Reconstructing past variations of the global mean surface temperature is used to characterise the Earth system response to perturbations as well as validate Earth system simulations. Reconstructing GMST beyond the instrumental period relies on algorithms aggregating local proxy temperature records.

Perhaps this can be written more clearer, because GMST from the instrumental period also requires an aggregation algorithm due to the non-uniform spatial distribution of thermometers on the planet.

Lines 4-5: Here, we propose to establish standards for the evaluation of the performance of such reconstruction algorithms.

In my opinion the above text does not correctly characterise the work. Firstly, in this study you are essentially comparing palaeoclimate data (from the Snyder approach) to palaeoclimate model output, so it is not strictly possible to independently evaluate either the data or the model output. We don't know if either the data or the model is correct, or if both are correct, or if neither are correct. Perhaps a more accurate statement that characterises the work would be that you "investigate the the level of agreement between data and model", which is of course a valuable exercise. As for

"establishing standards", I don't agree with this statement. The authors put forth an interesting approach for a data-model comparison, but I don't know why it should be a standard. Other authors may use different approaches, would their approach then be non-standard?

Lines 10-11: We find the algorithm to be able to reconstruct timescales longer than 4 kyr over the last 25 kyr. However, beyond 40 kyr BP,age uncertainty limits the algorithm capability to timescales longer than 15 kyr.

Do you mean: "temporal resolution of the algorithm is limited to 4 kyr for the last 25 kyr"? (This assertion is of course based on the assumption that the synthetic palaeoclimate data represents the truth, or at least a more complete effort to quantify of the truth).

In Lines 75 to 85 the authors explain why they use the PalMod database of proxy data as opposed to the original dataset of Snyder. I guess the main reason is to us a newer dataset with more data than the Snyder one (2016 was 8 years ago now). I don't agree with some of the reasoning given regarding superior chronological control in PalMod. Yes, Bacon was used to construct 14C + d18O tuning age models, but Bacon is only as good as what is put into it and is known to underestimate the total age population contained in multi-specimen discrete-depth sediment samples from deep-sea sediment (Bacon was originally developed for lacustrine sediment). Age-depth points based on d18O here are base based on visual matching benthic d18O data to regional benthic d18O curves (Lisiecki and Stern, 2016), which are themselves dated based on visual matching benthic d18O data to Greenland and speleothems, so we can say the age is "double tuned", with double potential interpretation error based on visual matching. Furthermore, there is a certain assumption of global synchronicity between all these records when tuning. Such approaches seems to be pretty standard in many palaeoclimate papers, so I don't wish to single out the authors in this case, but in a paper that seeks to quantify all sources of error, I believe these potential pitfalls should be pointed out clearly.

As for 14C, 14C in age-depth models in bioturbated archives can display large age-depth artefacts during periods of highly dynamic D14C (such as the last deglaciation) due to forams with very different 14C activities being combined into the same sample (Lougheed et al, 2022; https://doi.org/10.5194/gchron-2-17-2020). This uncertainty could possibly be also mentioned and considered...

Greenhouse gas concentrations are prescribed following the measurements from the EPICA DOME C ice core (Lüthi et al., 2008),

Last glacial EPICA Dome C is largely dated by making assumptions about local temperature and obliquity (Parrenin, et al., 2007; doi: 10.5194/cp-3-485-2007). Therefore, the greenhouse gas data from EPICA Dome C is unfortunately not independent of palaeoclimate assumptions, with consequences for LOVECLIM runs forced by EPICA Dome C greenhouse gases.

Comments on section **3.2.1 Construction of pseudo-proxies with sedproxy**

Please mention and cite the bioturbation model that is used within sedproxy (Berger and Heath)

One of the major issues with deep-sea sediment records based on multispecimen foram samples is the interaction between bioturbation and temporal abundance (e.g. Löwemark et al 2008; 10.1016/j.margeo.2008.10.005), meaning that centuries and/or millennia of high abundance are overrepresented in the sediment archive. Does the approach using FAME correctly account for granular changes in foram species abundance as a fraction of the total sediment flux?

Comments on section **3.2.2 Specification of the pseudo-proxy experiments**

I think it is good that the authors try to do many experimental setups, but at the same time, the tradeoff is that I as a reader I start to lose the ability here to keep track of them all. Admittedly, I have not slept properly in recent years.

Comments on section **4 Specification of the pseudo-proxy experiments**
It is unclear to me here what is being discussed here and shown in Fig 3.

If I am correct, the aim of this study is to compare the Snyder algorithm GMST to the pseudoproxy GMST, right?

Fig 3 just shows the ensemble results from the pseudoproxy GMST, right? Not the Snyder algorithm. Yet the text in the paragraph around line 300 refers to Figure 3 in the context of the "ability of the GMST reconstruction alogrithm".

I think I figured out now what is in Fig 3… for example in Fig 3C, is the black line the Snyder approach and the red line the ensemble mean from pseudoproxy approach? (Or is it the other way around?) So essentially the the text is referring to the agreement between the red and the black line (for Fig 3C).

Perhaps clearer and more consistent language should be used here, because my main issue is figuring out what is what ("reconstructed GMST" and "simulated GMST"). Use "PPE GMST" and "S16 GMST" throughout the text and the caption here? Although this risks introducing more acronyms.

Line 316: "One of the most important contributors to the uncertainty estimate is the scaling factor."

Again here, when referring to the "uncertainty estimate" it is unclear at first reading if you are referring to the Snyder approach or to your PPE approach.

Line 372: We also estimate an upper bound to the effect of seasonality bias on the pseudo-proxy reconstruction. If all records were not recording the mean annual temperature, but that of the warmest month, it would generate biases up to 0.75 K. This bias mostly follows the northern hemisphere insolation curve:

What is the northern hemisphere insolation curve? Be more specific.

Line 375: These orbital-scale biases can significantly impact the evaluation of orbital timescale variations

What is orbital-scale and orbital timescale? Obliquity, eccentricity and longitude of perihelion are changing as I am typing this sentence. Perhaps be more specific as to the exact timescales you are referring to. Half an obliquity cycle? (20.5 kyr?)

This warm season bias has, for example, been one of the main hypothesised reasons for the model-data discrepancies during the early Holocene (Liu et al., 2014; Marsicek et al., 2018; Bova et al., 2021), although there is also evidence for cold season bias for other periods and species (Steinke et al., 2008; Timmermann et al., 2014).

Bova et al essentially detrend their data for general precession, which is one of the major drivers of Quaternary global climate.

Line 560: First, the age uncertainty is the limiting factor preventing the reconstruction of multi-millennial timescale beyond 30 kyr BP, and the main focus should be put on reducing this uncertainty (e.g. Waelbroeck et al., 2019; Peeters et al., 2023).

I think all Earth Scientists would like to see reduced age uncertainty in data, I'm not sure if those two papers were the first to point it out. In particular the Waelbroeck paper concentrates on better implementing age uncertainty… in some cases this actually *increased* the age uncertainty over the original datasets. So I would say the main focus should be in quantifying age uncertainty. If it gets reduced then that's a bonus.

Comments on section **5.3 Improvement of the algorithm**

Once again, here the paper is being framed as a way to evaluate an algorithm, by comparing to how it compares to pseudoproxy data developed from climate model runs. This assumes that the latter represents the "truth" and that the algorithm must be evaluated against this truth.

---

## Author Comment (AC2)

Short summary on anticipated changes to the manuscript:
- We will update the title to "Testing the reliability of global surface temperature reconstructions of the last glacial cycle using pseudo-proxy experiments"
- We will clarify in the data section that we only use the records metadata from the proxy database, and not the SST estimates.
- We will improve the readability of the plots.
- We will enhance the paper with the points the reviewer suggested.

Response to comments from Kaustubh Thirumalai

Major Comment 1

***Bioturbation mixing depth parameterization****: The authors conclude that "Our results show also the existence of a trade-off between the inclusion of a large number of records, which overall reduces the uncertainty, and of only the highest quality records (low age uncertainty, high accumulation rate, high resolution), which improves the reconstruction of the short timescale" where the latter refers to reconstructions being relatively free from bioturbation- and age-model-related smoothening to sufficiently preserve short-timescale (or multicentennial-to-millennial) climatic signals. The manuscript (Table 1 and Figure 2 caption) suggests that the authors used a constant (presumably?) sediment mixed-layer depth (Table 1 only indicates 'bioturbation' and would benefit from more explicit details about what is being parameterized) of 10 cm. In my opinion, and according to my cursory assessment of global bioturbation rates (I realize that argue otherwise—but I'd like to see the numbers) of the data presented in Teal et al. (2010)—10 cm is far too high for average global mixing depths, particularly given J20's bias towards tropical and near-coastal proxy locations. I would like to see how a value of, for example, 5 cm would perform for the Full PP and related experiments. I understand that the authors have attempted to parse the sensitivity of 'age uncertainty' versus 'bioturbation' and other associated parameters in their analysis, but this does not address the entire hierarchy of choices with a lower bioturbation rate. If feasible, I'd recommend that the authors perform such an experiment (Full PP with reduced bioturbation rate) and check whether more high-frequency information is retained in the associated spectra.*

This is a valid point, we did not further investigate the impact of different bioturbation depth on the signal. We agree that, since Boudreau 1998 evaluated mixing depth at about 10cm, more recent studies have suggested lower mean values (Teal et al. 2008, Zhang et al. 2024). One justification to our use of a conservative estimate is to account for temporal variability, and the possible occurrence of higher values. Even if episodes of deep bioturbation last a short time, they may have a large impact on the signal if the sedimentation rate is low. A second justification is the weak impact of smaller depths on our result as shown in the Fig.1 below. In this figure we provide a comparison of pseudo-proxy experiments with 10 cm and 5 cm bioturbation layers. Looking first at the effect of bioturbation only, we find as expected that the drop in the coherence occurs at half the timescale than if we use a 50% smaller bioturbation depth. However, if we consider all factors (Full PP), the change does not sensibly affect our results. This is because bioturbation is never the sole main factor explaining the drop in coherence. It is nevertheless worth noting that, in the case of the last 25 kyr (Panel B and C), bioturbation does not seem to play a relevant role in the smoothing of the signal any more when the mixing depth is reduced to 5cm. We will add this information to the manuscript.

[Figure]

*Figure 1: Impact of bioturbation rates on the coherence of the signal. We test two mixed-layer depths: 10 cm (light colours, same data as in the manuscript in Fig.8) and 5 cm (dark colours).*

Major Comment 2
***Clarity regarding the use+utility of J20****: Unless I am mistaken (which is highly likely), the authors do not actually use or show the GMST calculated (using any algorithm) from the reconstructions collated in J20. They merely use the metadata of proxy parameters within J20 as a framework for their pseudo-proxy experiments. The clarity of the manuscript would be improved if the authors were more upfront about this aspect in their abstract and introduction. On the other hand, this also leads me to question this omission as potential comparisons between reconstructed (from data), simulated, and resampled GMST calculations would be highly interesting. However, I recognize that this may be beyond the scope of this work—accordingly, I feel that the authors should preface this aspect and consider using the term metadata in their abstract and text. I think that the title of the manuscript should include/reflect 'sensitivity/uncertainty experiments' and/or 'pseudo-proxy experiments' because in its current form, without comparing simulated and actually reconstructed GMST, I do not think the authors can claim to test the 'reliability the reliability of global surface temperature reconstructions of the last glacial cycle'. Rather, they are testing the reliability of the methodologies used to create global mean reconstructions… hence I feel that a title revision is needed.*

- You are correct, we only use the metadata of the J20 dataset, not the actual measurements. We will clarify this on the manuscript. However, producing and analysing the GMST reconstruction based on the J20 data is out of scope for this manuscript.
- We argue that the pseudo-proxy experiments provide a more robust methodology to test the reliability of the reconstructions than model-data comparisons. In particular, directly comparing model outputs and proxy data suffers from the same limitations as the one we face when extrapolating our quantitative results from model data to real reconstructions (e.g. the LGM bias). However, our evaluation of the reconstruction algorithm still provides further insights, in particular regarding which timescale can be reliably reconstructed. We will add this argument in the introduction. In addition, given that all our study revolves around PPE, it is worth adding it to the title.

Major Comment 3
**Inclusion of 'Full PP at random locations':** *It appears that the authors do not show any results from this experiment, yet it plays an important role in their analysis (see, e.g., Lines 335–340: "In addition, location resampling and latitude band configurations, which aim to account for it, are not large enough to cover the bias. Yet, the pseudo-proxy experiments using random proxy locations can reproduce the simulated MSST.") I recommend that the authors make a plot showing results from this experiment and to be more quantitative/precise regarding the ability or lack thereof of these simulations to reproduce simulated mean SSTs.*

We designed the experiment "Full PP at random locations" mostly to investigate the impact of proxy numbers on the reconstructions (L. 350-356). We realised later that they could also be used to study the effect of locations biases. The respective timeseries are shown below in Fig.2 . We can see that the increased number of records has only a small impact on the mean reconstructed values, and that these are very close to a smoothed version of the simulated GMST. However, the increase in the number of proxy time series reduces noticeably the confidence interval.

[Figure]

*Figure 2: Reconstructions based on the experiments "Full PP at random locations", for various number of records. The reconstructions (colours) are compared to the simulated GMST (black). The shading correspond to the 90% CI.*

***Attribution of a specific set of J20 locations as a significant bias:*** *Based on the last point, the authors state that "Therefore, the bias is caused by the specific set of locations in the J20 dataset: there is an over-representation of regions with strong LGM cooling". Whereas this assessment may be accurate, I do not find the regions that the authors identify to be a convincing explanation (e.g., NW Atlantic/Kurushio extension)—instead, it seems to me that this is a bias related to the relative proximity of core locations to continents—where land-based cooling strongly impacts these sites as opposed to open-ocean marine conditions. Is it possible for the authors to combine inferences from the 'Full PP at random locations' or another sensitivity experiment to test this possibility?*

We show below the map of LGM (19-23kyr) temperature anomaly as used to compute the pseudo-proxies for the 5 simulations, with the location of records with data in both the last 5kyr and the LGM. There is no clear specific cooling pattern at the proximity of the continents in the simulations. By contrast, there are large differences in and between the North Pacific and the North Atlantic, where the sampling of location is quite heterogeneous. In the text, we point in particular to undersampled regions where the cooling is weaker, hence explaining the cold bias in the reconstructions. Note that we do not rule out a cooling bias due to the proximity of the continents in the proxy reconstructions. The simulations we use simply do not show such an effect.

[Figure]

*Figure 3: LGM temperature anomaly for the 5 simulations. These are the temperatures used in the pseudo-proxies (SST or ts with the proper land mask). The crosses correspond to the record locations with at least one value during the LGM (19-23 kyr) and one during the reference time period (0-5kyr).*

Minor Suggestions
*Line 81: "…needed to develop our evaluation standards." Please rephrase.*
- This sub-sentence is not needed and indeed a bit confusing. We will remove it.

*Lines 91–93: Are there only 7 (112–105) unspecified datasets? Figure 1 says otherwise—please clarify.*
- No there are 112. 105(annual)+112(unspecified)+48(pseudo-annual) = 265 (total)

*Lines 162–163: Please consider adding more information to contextualize why the following steps are being performed. This would be a great spot to clarify the involvement/extent of usage (or lack thereof) of the actual records within J20.*
- What is used from J20 is more relevant for the construction of the pseudo-proxy, and we prefer to add it in the data section. As for here, we will change the sentence to be more explicit: "We decide to adapt the S16 algorithm to make full use of the J20 dataset (age ensembles, records with no data in the last 5kyr), and improve the uncertainty quantification by the algorithm."

*Lines 236–237: 0.26 K and 0.23 K seem to be exceedingly low values for analytical uncertainty. Does this take into account sampling uncertainty (see e.g. Thirumalai et al. 2013) which is the uncertainty that foraminiferal shells (with lifespans of a ~month) would have grown at different points of time within the sampling interval, and thus will have uncertainty in reconstructing the 'interval mean'? If sedproxy takes this into account, it would be good to mention this aspect.*
- Sampling uncertainty is applied for Mg/Ca as explained in the paragraph just above (L.228-235). The values are otherwise justified in Dolman and Laepple 2018.

*Lines 245–246: Have you considered performing a depth-sensitivity test? i.e. instead of the uppermost grid location, what about the integration of the top 50 m—which is a more realistic scenario for the proxy integration of temperature signals for the chosen sensors. Perhaps this also might explain the cool bias?*
- The main limitation to analyse the sensitivity to depth is data availability. Using a constant depth would most likely lead to slightly smaller amplitude of variation and to warmer LGM anomaly. This does not constitute an explanation to the cool bias however, but instead add a layer of uncertainty. Change in habitat depth through time can also be even more impactful, as we mention in the discussion / outlook of the paper.

*General comment on Figures: Where there are many lines depicted in figures, it is very difficult to parse the colors of each line (especially on the spectral plots) and to associate them with the legend. I would strongly consider using a different backdrop color or thicker lines with a different subplot layout to more cleary delineate results from various experiments.*
- Thank you for the feedback, we will increase the thickness of the lines, particularly in the legend, and test if the plots can be split in two.

*Discussion and Lines 545–555: The authors should consider discussing the values of **scaling utilized in Clark et al. (2024)** and how they fit within this portion of the discussion.*
-> We present below a scatter plot of GMST VS GSST in the simulation we used, including Snyder 2016 and Clark et al. 2024, with uncertainty. In short, the two are very similar, but the uncertainty range behave differently. Hence, it doesn't seem that the polynomial fit of Clark et al 2024 brings a better agreement. In addition, we would argue that neither a purely additive noise as in Clark et al 2024 or a purely multiplicative noise as in Snyder 2016 are able to capture the uncertainty correctly. Part of the issue in Clark et al. 2024 could be that they only use CESM-based simulations for the colder than PI part of the fit, which undersample the uncertainty. We will add a couple of sentences in the paragraph about the new fit from Clark et al. 2024, thank you for pointing it out.

[Figure]

*Figure 4: Comparison of the scaling between GMST and GSST anomaly. The dots correspond to the various simulations used in the study, with the colour indicating the time period. The dark orange line corresponds to the scaling used in Snyder 2016, and the shading to 2 standard deviations. The blue line is the scaling used in Clark et al. 2024, with the shading also corresponding to 2 times the standard deviation.*

---

## Author Comment (AC3)

Short summary on anticipated changes to the manuscript:
- We will clarify in the abstract, introduction and methodology, that we focus on evaluating a reconstruction algorithm, and for that only use the metadata from the J20 dataset to compute pseudo-proxies.
- We will enhance the discussion with the points the reviewer suggested and correct the various imprecisions that were noticed.
- We will add the flowchart as suggested in the second reply.

Point by point response to Bryan Lougheed:

*Lines 1-3: Perhaps this can be written more clearer, because GMST from the instrumental period also requires an aggregation algorithm due to the non-uniform spatial distribution of thermometers on the planet.*
- "Beyond the instrumental period, reconstructions rely on local proxy temperature records and algorithms aggregating these records."

*Lines 4-5: In my opinion the above text does not correctly characterise the work. Firstly, in this study you are essentially comparing palaeoclimate data (from the Snyder approach) to palaeoclimate model output, so it is not strictly possible to independently evaluate either the data or the model output. We don't know if either the data or the model is correct, or if both are correct, or if neither are correct. Perhaps a more accurate statement that characterises the work would be that you "investigate the the level of agreement between data and model", which is of course a valuable exercise. As for "establishing standards", I don't agree with this statement. The authors put forth an interesting approach for a data-model comparison, but I don't know why it should be a standard. Other authors may use different approaches, would their approach then be non-standard?*
- The misunderstanding has been clarified in the previous short reply (https://doi.org/10.5194/egusphere-2024-1387-AC1). In short, we use model data to evaluate the reconstruction algorithm. Given the lack of evaluation of reconstruction algorithms, we propose the methodology we introduce here as a standard.

*Lines 10-11: Do you mean: "temporal resolution of the algorithm is limited to 4 kyr for the last 25 kyr"? (This assertion is of course based on the assumption that the synthetic palaeoclimate data represents the truth, or at least a more complete effort to quantify of the truth).*
- We will rephrase to: "The reconstruction based on the J20 database and the S16 algorithm is reliable for timescales above 4kyr during the last 25kyr.", We prefer the term "timescale" to that of "temporal resolution", because the later could refer to the timestep of the reconstruction, which is fixed to 100 years here.

*In Lines 75 to 85 the authors explain why they use the PalMod database of proxy data as opposed to the original dataset of Snyder. I guess the main reason is to us a newer dataset with more data than the Snyder one (2016 was 8 years ago now). I don't agree with some of the reasoning given regarding superior chronological control in PalMod. Yes, Bacon was used to construct 14C + d18O tuning age models, but Bacon is only as good as what is put into it and is known to underestimate the total age population contained in multi-specimen discrete-depth sediment samples from deepsea sediment (Bacon was originally developed for lacustrine sediment). Age-depth points based on d18O here are base based on visual matching benthic d18O data to regional benthic d18O curves (Lisiecki and Stern, 2016), which are themselves dated based on visual matching benthic d18O data to Greenland and speleothems, so we can say the age is "double tuned", with double potential interpretation error based on visual matching. Furthermore, there is a certain assumption of global synchronicity between all these records when tuning. Such approaches seems to be pretty standard in many palaeoclimate papers, so I don't wish to single out the authors in this case, but in a paper that seeks to quantify all sources of error, I believe these potential pitfalls should be pointed out*

*clearly. As for 14C, 14C in age-depth models in bioturbated archives can display large age-depth artefacts during periods of highly dynamic D14C (such as the last deglaciation) due to forams with very different 14C activities being combined into the same sample (Lougheed et al, 2022; https://doi.org/10.5194/gchron-2-17-2020). This uncertainty could possibly be also mentioned and considered...*

- We did not mean that the age models in J20 are superior to those in the S16 datasets. However, the availability of age ensembles for J20 facilitates a more sophisticated treatment of the effect of age uncertainty on the reconstruction, in comparison to the standard deviation used in S16.
- We agree with the reviewer that the age ensembles in J20 do not consider the errors arising from the visual matching, and that a synchronicity is indeed assumed, albeit only regionally. Age uncertainty is already the largest source of signal smoothing for the period where the matching is the most relevant. Hence, considering these errors does not qualitatively change our result, but would increase the timescale below which the reconstruction is not reliable. We will add a sentence in the discussion.

Finally, your point on bioturbation affecting age uncertainty is very interesting, and we will add it to the relevant discussion. With sedproxy, we only consider the smoothing effect of bioturbation but not how it impacts radiocarbon ages. To do this, a more complex pseudo-proxy algorithm would be needed, in particular, one that re-computes the age ensembles.

*Last glacial EPICA Dome C is largely dated by making assumptions about local temperature and obliquity (Parrenin, et al., 2007; doi: 10.5194/cp-3-485-2007). Therefore, the greenhouse gas data from EPICA Dome C is unfortunately not independent of palaeoclimate assumptions, with consequences for LOVECLIM runs forced by EPICA Dome C greenhouse gases.*

- The previous short reply should partially clarify this. We compare data from the same simulations (simulated GMST VS reconstructed GMST), so the comparison is not affected by the assumptions made on the data forcing the simulations.

Section 3.2.1
*Please mention and cite the bioturbation model that is used within sedproxy (Berger and Heath)*
- We will add it (Berger and Heath, 1968).

*One of the major issues with deep-sea sediment records based on multispecimen foram samples is the interaction between bioturbation and temporal abundance (e.g. Löwemark et al 2008; 10.1016/j.margeo.2008.10.005), meaning that centuries and/or millennia of high abundance are overrepresented in the sediment archive. Does the approach using FAME correctly account for granular changes in foram species abundance as a fraction of the total sediment flux?*
- You are correct, this is exactly what the approach does. The pseudo-proxy temperature is weighted towards the time and temperatures favouring a given species, not only seasonally but also through the bioturbated layer. We will correct the text accordingly.

Section 3.2.2
*I think it is good that the authors try to do many experimental setups, but at the same time, the tradeoff is that I as a reader I start to lose the ability here to keep track of them all. Admittedly, I have not slept properly in recent years.*
- We understand this could be difficult for the reader. We tried to make the text as readable as possible, while the table offers a quick overview of each experiment.

Section 4
*It is unclear to me here what is being discussed here and shown in Fig 3.*

- We clarified this in the short answer, and will add to the caption that the reconstructed GMST is based on the S16 algorithm, and that the simulated GMST is that of the respective climate simulations.

*If I am correct, the aim of this study is to compare the Snyder algorithm GMST to the pseudoproxy GMST, right?*
*Fig 3 just shows the ensemble results from the pseudoproxy GMST, right? Not the Snyder algorithm. Yet the text in the paragraph around line 300 refers to Figure 3 in the context of the "ability of the GMST reconstruction alogrithm".*
*I think I figured out now what is in Fig 3… for example in Fig 3C, is the black line the Snyder approach and the red line the ensemble mean from pseudoproxy approach? (Or is it the other way around?) So essentially the the text is referring to the agreement between the red and the black line (for Fig 3C).*
*Perhaps clearer and more consistent language should be used here, because my main issue is figuring out what is what ("reconstructed GMST" and "simulated GMST"). Use "PPE GMST" and "S16 GMST" throughout the text and the caption here? Although this risks introducing more acronyms.*
- These comments have been addressed in the short answer. We do not use the reconstruction in S16, only the algorithm, which we applied to the pseudo-proxy data computed from model data ("*reconstructed GMST*"). The *simulated GMST* is simply the area-weighted mean of the surface temperature field produced by the simulation.

*Line 316: Again here, when referring to the "uncertainty estimate" it is unclear at first reading if you are referring to the Snyder approach or to your PPE approach.*
- Here, the uncertainty is the one estimated by the Snyder algorithm. It can be split into the 5 components for either PPE reconstructions or real reconstructions (not performed here). However, the validity of the uncertainty estimate can only be determined in a PPE experiment, where the truth is known.

*Line 372: What is the northern hemisphere insolation curve? Be more specific.*
- Summer solstice insolation at 65°N, we will add it to the text.

*Line 375: What is orbital-scale and orbital timescale? Obliquity, eccentricity and longitude of perihelion are changing as I am typing this sentence. Perhaps be more specific as to the exact timescales you are referring to. Half an obliquity cycle? (20.5 kyr?)*
- We mean timescales > 10kyr., i.e., at least half a precession cycle. We will change this in the text.

*Bova et al essentially detrend their data for general precession, which is one of the major drivers of Quaternary global climate.*
- We agree

*Line 560: I think all Earth Scientists would like to see reduced age uncertainty in data, I'm not sure if those two papers were the first to point it out. In particular the Waelbroeck paper concentrates on better implementing age uncertainty… in some cases this actually increased the age uncertainty over the original datasets. So I would say the main focus should be in quantifying age uncertainty. If it gets reduced then that's a bonus.*
- We agree, the sentence is not very well formulated. We will rather discuss the need to better understand the age uncertainty in order to better constraint it. That will better fit the work by Waelbroeck et al., 2019, and we can include one of the paper you suggested here (Lougheed et al, 2022).

Section 5.3

*Once again, here the paper is being framed as a way to evaluate an algorithm, by comparing to how it compares to pseudoproxy data developed from climate model runs. This assumes that the latter represents the "truth" and that the algorithm must be evaluated against this truth.*
- This has been clarified in the short answer. We do evaluate the algorithm, and thanks to the PPE framework, we can compare the reconstructed GMST to the simulated GMST from the same simulation, which in that case represents the target.

---

## Author Response (AR1)

**Reponse to  Bryan Lougheed:**

To the general comment:

We clarified throughout the text that we only used the metadata of J20 and did not compute the reconstruction based on the real proxy data. We also reorganised the introduction to put emphasis on the evaluation rather than on the specific dataset and algorithm.

Added text: "*the metadata of the synthesis*" (L8)

Updated introduction (L40-65)
*To our knowledge, a proper evaluation of such reconstruction methods hasn't been performed, which limits their use for model evaluations. In particular, we note several open questions regarding the quality of GMST reconstructions of and beyond the LGC:*
*– How reliable are GMST reconstructions of the LGC?*
*– Does the non-uniform spatiotemporal distributions of proxy samples lead to biased GMST reconstructions?*
*– Which sources of uncertainty impact reconstructions, and in what way?*
*– What is the shortest timescale on which amplitudes and timing of GMST variations can be accurately reconstructed, for a given algorithm and dataset?*
*– What are the sources of the loss of accuracy on short timescales?*
*– Which limiting factors should be prioritised for improving the GMST reconstruction quality (uncertainty, resolution)?*

*This study tests a reconstruction method similar to the one published in Snyder (2016, hereafter S16). This algorithm was used to compute a GMST reconstruction of the Pleistocene based on SST proxy records to investigate glacial cycles. It is 2robust enough to tackle and account for the non-stationarity of climate states and the sparsity of the records. Compared to other algorithms proposed to reconstruct GMST on glacial-interglacial timescales, it quantifies uncertainties more comprehensively and has a limited reliance on model output. Therefore, we use it as the basis of our work, although most of the findings will likely hold for other algorithms using similar principles for the reconstruction.*

*This study does not provide a new reconstruction, but an evaluation method based on pseudo-proxy experiments (PPEs) and transient climate simulations. PPEs enable the test of algorithms in controlled, idealised environments, where the underlying truth is known (i.e. the simulation outputs). They rely on pseudo-proxies that are synthetic time series imitating proxy records, based on the spatiotemporal climate state of the simulation. The computation of pseudo-proxies is performed using a proxy system model (sedproxy, Dolman and Laepple, 2018), which simulates the processes from the fixation of the climate-dependent quantity to the measurement of the proxy in the lab, including the entailed uncertainties. To create realistic pseudo-proxies, we use the metadata of a recently published database: the PalMod 130k marine palaeoclimate data synthesis (Jonkers et al., 2020, hereafter J20). This database provides numerous proxy records of near-surface sea temperature from marine sediments. It is metadata-rich, which enables us to test the impact of various uncertainty sources on the reconstruction, and whether these are well accounted for in the algorithm's uncertainty estimates.*

Also: "*Finally, we discuss the performance of the algorithm when applied to the J20 database, and future avenues to improve GMST reconstructions.*" (L68)

Data section:

- L76 "*We use various metadata of this database: core locations, proxy types, species (for M g/Ca) sedimentation rates, and age ensembles.*"
- L83: "*We do not use the temperature records in J20, as our study does not aim to present a new GMST reconstruction.*"

We also added a flowchart (Fig. 3) as suggested by the reviewer, and a reference to it in the text (L 194).

Point by point response to Bryan Lougheed:

**Lines 1-3: Perhaps this can be written more clearer, because GMST from the instrumental period also requires an aggregation algorithm due to the non-uniform spatial distribution of thermometers on the planet.**
- "*Beyond the instrumental period, reconstructions rely on local proxy temperature records and algorithms aggregating these records.*"

**Lines 4-5: In my opinion the above text does not correctly characterise the work. Firstly, in this study you are essentially comparing palaeoclimate data (from the Snyder approach) to palaeoclimate model output, so it is not strictly possible to independently evaluate either the data or the model output. We don't know if either the data or the model is correct, or if both are correct, or if neither are correct. Perhaps a more accurate statement that characterises the work would be that you "investigate the the level of agreement between data and model", which is of course a valuable exercise. As for "establishing standards", I don't agree with this statement. The authors put forth an interesting approach for a data-model comparison, but I don't know why it should be a standard. Other authors may use different approaches, would their approach then be non-standard?**
- See response to general comment.

**Lines 10-11: Do you mean: "temporal resolution of the algorithm is limited to 4 kyr for the last 25 kyr"? (This assertion is of course based on the assumption that the synthetic palaeoclimate data represents the truth, or at least a more complete effort to quantify of the truth).**
- We rephrased to: "*Given the dataset and the algorithm, we find that the reconstruction is reliable for timescales above 4 kyr during the last 25 kyr.*", We prefer the term "timescale" to that of "temporal resolution", because the later could refer to the timestep of the reconstruction, which is fixed to 100 years here.

**In Lines 75 to 85 the authors explain why they use the PalMod database of proxy data as opposed to the original dataset of Snyder. I guess the main reason is to us a newer dataset with more data than the Snyder one (2016 was 8 years ago now). I don't agree with some of the reasoning given regarding superior chronological control in PalMod. Yes, Bacon was used to construct 14C + d18O tuning age models, but Bacon is only as good as what is put into it and is known to underestimate the total age population contained in multi-specimen discrete-depth sediment samples from deepsea sediment (Bacon was originally developed for lacustrine sediment). Age-depth points based on d18O here are base based on visual matching benthic d18O data to regional benthic d18O curves (Lisiecki and Stern, 2016), which are themselves dated based on visual matching benthic d18O data to Greenland and speleothems, so we can say the age is "double tuned", with double potential interpretation error based on visual matching. Furthermore, there is a certain assumption of global synchronicity between all these records when tuning. Such approaches seems to be pretty standard in many palaeoclimate papers, so I don't wish to single out the authors in this case, but in a paper that**

**seeks to quantify all sources of error, I believe these potential pitfalls should be pointed out clearly. As for 14C, 14C in age-depth models in bioturbated archives can display large age-depth artefacts during periods of highly dynamic D14C (such as the last deglaciation) due to forams with very different 14C activities being combined into the same sample (Lougheed et al, 2022; https://doi.org/10.5194/gchron-2-17-2020). This uncertainty could possibly be also mentioned and considered...**

- We did not mean that the age models in J20 are superior to those in the S16 datasets. However, the availability of age ensembles for J20 facilitates a more sophisticated treatment of the effect of age uncertainty on the reconstruction, in comparison to the standard deviation used in S16.

- We agree with the reviewer that the age ensembles in J20 do not consider the errors arising from the visual matching, and that a synchronicity is indeed assumed, albeit only regionally. Age uncertainty is already the largest source of signal smoothing for the period where the matching is the most relevant. Hence, considering these errors does not qualitatively change our result, but would increase the timescale below which the reconstruction is not reliable.

- Finally, your point on bioturbation affecting age uncertainty is very interesting, and we added it to the relevant discussion. With sedproxy, we only consider the smoothing effect of bioturbation but not how it impacts radiocarbon ages. To do this, a more complex pseudo-proxy algorithm would be needed, in particular, one that re-computes the age ensembles.

Added: "*or considering the impact of bioturbation on age model (Lougheed et al 2022)*" (L566)

**Last glacial EPICA Dome C is largely dated by making assumptions about local temperature and obliquity (Parrenin, et al., 2007; doi: 10.5194/cp-3-485-2007). Therefore, the greenhouse gas data from EPICA Dome C is unfortunately not independent of palaeoclimate assumptions, with consequences for LOVECLIM runs forced by EPICA Dome C greenhouse gases.**

- See response to general comment. We compare data from the same simulations (simulated GMST VS reconstructed GMST), so the comparison is not affected by the assumptions made on the data forcing the simulations.

**Section 3.2.1**
**Please mention and cite the bioturbation model that is used within sedproxy (Berger and Heath)**

- We added it

**One of the major issues with deep-sea sediment records based on multispecimen foram samples is the interaction between bioturbation and temporal abundance (e.g. Löwemark et al 2008; 10.1016/j.margeo.2008.10.005), meaning that centuries and/or millennia of high abundance are overrepresented in the sediment archive. Does the approach using FAME correctly account for granular changes in foram species abundance as a fraction of the total sediment flux?**

- You are correct, this is exactly what the approach does. The pseudo-proxy temperature is weighted towards the time and temperatures favouring a given species, not only seasonally but also through the bioturbated layer.

Replaced "*seasonal preferences*" by "*overrepresentation of high abundance periods, such as particular seasons*"

**Section 3.2.2**
*I think it is good that the authors try to do many experimental setups, but at the same time, the tradeoff is that I as a reader I start to lose the ability here to keep track of them all. Admittedly, I have not slept properly in recent years.*

**-** We understand this could be difficult for the reader. We tried to make the text as readable as possible, while the table offers a quick overview of each experiment.

Section 4

**It is unclear to me here what is being discussed here and shown in Fig 3.**
- See response to general comment.
The caption of Fig.4 now reads: "*Comparison of the simulated GMST (black lines) and the corresponding reconstructed GMST (coloured line) for each simulation*."

**If I am correct, the aiComparison of the simulated GMST (black lines) and the corresponding reconstructed GMST (coloured line) for each simulation.m of this study is to compare the Snyder algorithm GMST to the pseudoproxy GMST, right?**
**Fig 3 just shows the ensemble results from the pseudoproxy GMST, right? Not the Snyder algorithm. Yet the text in the paragraph around line 300 refers to Figure 3 in the context of the "ability of the GMST reconstruction alogrithm".**
**I think I figured out now what is in Fig 3… for example in Fig 3C, is the black line the Snyder approach and the red line the ensemble mean from pseudoproxy approach? (Or is it the other way around?) So essentially the the text is referring to the agreement between the red and the black line (for Fig 3C).**
**Perhaps clearer and more consistent language should be used here, because my main issue is figuring out what is what ("reconstructed GMST" and "simulated GMST"). Use "PPE GMST" and "S16 GMST" throughout the text and the caption here? Although this risks introducing more acronyms.**
- See response to general comment. We do not use the reconstruction in S16, only the algorithm, which we applied to the pseudo-proxy data computed from model data (" *reconstructed GMST*"). The *simulated GMST* is simply the area-weighted mean of the surface temperature field produced by the simulation.

**Line 316: Again here, when referring to the "uncertainty estimate" it is unclear at first reading if you are referring to the Snyder approach or to your PPE approach.**
- See response to general comment. Here, the uncertainty is the one estimated by the Snyder algorithm. It can be split into the 5 components for either PPE reconstructions or real reconstructions (not performed here). However, the validity of the uncertainty estimate can only be determined in a PPE experiment, where the truth is known.
Text added:
- L312: "*Note that the uncertainty estimates do not depend on the uncertainty sources accounted for when creating the pseudo-proxies*."
- L328: "*We can use the PPEs that only include one uncertainty source, and compare it to the estimates from the algorithm*."

**Line 372: What is the northern hemisphere insolation curve? Be more specific.**
Added "*Summer solstice insolation at 65°N*"

**Line 375: What is orbital-scale and orbital timescale? Obliquity, eccentricity and longitude of perihelion are changing as I am typing this sentence. Perhaps be more specific as to the exact timescales you are referring to. Half an obliquity cycle? (20.5 kyr?)**
Replaced "*orbital-scale*" by "*precession-scale*"

**Bova et al essentially detrend their data for general precession, which is one of the major drivers of Quaternary global climate.**
- We agree

**Line 560: I think all Earth Scientists would like to see reduced age uncertainty in data, I'm not sure if those two papers were the first to point it out. In particular the Waelbroeck paper concentrates on better implementing age uncertainty… in some cases this actually increased the age uncertainty over the original datasets. So I would say the main focus should be in quantifying age uncertainty. If it gets reduced then that's a bonus.**

- We agree, the sentence is not very well formulated. We will rather discuss the need to better understand the age uncertainty in order to better constraint it. That will better fit the work by Waelbroeck et al., 2019, and we can include one of the paper you suggested here (Lougheed et al, 2022).

Corrected text: "*Progress has been made for example using visual matching and quantifying the associated uncertainty or considering the impact of bioturbation on age model (e.g. Waelbroeck et al., 2019; Lougheed, 2022; Peeters et al., 2023)*"

Section 5.3

**Once again, here the paper is being framed as a way to evaluate an algorithm, by comparing to how it compares to pseudoproxy data developed from climate model runs. This assumes that the latter represents the "truth" and that the algorithm must be evaluated against this truth.**

- See response to general comment. We do evaluate the algorithm, and thanks to the PPE framework, we can compare the reconstructed GMST to the simulated GMST from the same simulation, which in that case represents the target.

############################################################################################

**Response to comments from Kaustubh Thirumalai**

Major Comment 1

**Bioturbation mixing depth parameterization: The authors conclude that "Our results show also the existence of a trade-off between the inclusion of a large number of records, which overall reduces the uncertainty, and of only the highest quality records (low age uncertainty, high accumulation rate, high resolution), which improves the reconstruction of the short timescale" where the latter refers to reconstructions being relatively free from bioturbation- and age-model-related smoothening to sufficiently preserve short-timescale (or multicentennial-to-millennial) climatic signals. The manuscript (Table 1 and Figure 2 caption) suggests that the authors used a constant (presumably?) sediment mixed-layer depth (Table 1 only indicates 'bioturbation' and would benefit from more explicit details about what is being parameterized) of 10 cm. In my opinion, and according to my cursory assessment of global bioturbation rates (I realize that   argue otherwise—but I'd like to see the numbers) of the data presented in Teal et al. (2010)—10 cm is far too high for average global mixing depths, particularly given J20's bias towards tropical and near-coastal proxy locations. I would like to see how a value of, for example, 5 cm would perform for the Full PP and related experiments. I understand that the authors have attempted to parse the sensitivity of 'age uncertainty' versus 'bioturbation' and other associated parameters in their analysis, but this does not address the entire hierarchy of choices with a lower bioturbation rate. If feasible, I'd recommend that the authors perform such an experiment (Full PP with reduced bioturbation rate) and check whether more high-frequency information is retained in the associated spectra.**

This is a valid point, we did not further investigate the impact of different bioturbation depth on the signal. We agree that, since Boudreau 1998 evaluated mixing depth at about 10cm, more recent studies have suggested lower mean values (Teal et al. 2008, Zhang et al. 2024). One justification to our use of a conservative estimate is to account for temporal variability, and the possible occurrence of higher values. Even if episodes of deep bioturbation last a short time, they may have a large impact on the signal if the sedimentation rate is low. A second justification is the weak impact of smaller depths on our result as shown in the Fig.S2 In this figure we provide a comparison of pseudo-proxy experiments with 10 cm and 5 cm bioturbation layers. Looking first at the effect of bioturbation only, we find as expected that the drop in the coherence occurs at half the timescale than if we use a 50% smaller bioturbation depth. However, if we consider all factors (Full PP), the change does not sensibly affect our results. This is because bioturbation is never the sole main factor explaining the drop in coherence. It is nevertheless worth noting that, in the case of the last 25 kyr (Panel B and C), bioturbation does not seem to play a relevant role in the smoothing of the signal any more when the mixing depth is reduced to 5cm. We will add this information to the manuscript.

Added Fig S2.
Added text:
- Table 1 caption: " *The parameter for the bioturbation is the bioturbated layer width*. "
- L 225: "*While other studies have suggested lower values (Teal et al., 2008; Zhang et al., 2024), we consider this a conservative approach to our analysis*."
- L465: "*In addition, using a smaller bioturbation depth further limits the influence of bioturbation (Fig. S2)*."

**Major Comment 2**
**Clarity regarding the use+utility of J20: Unless I am mistaken (which is highly likely), the authors do not actually use or show the GMST calculated (using any algorithm) from the reconstructions collated in J20. They merely use the metadata of proxy parameters within J20 as a framework for their pseudo-proxy experiments. The clarity of the manuscript would be improved if the authors were more upfront about this aspect in their abstract and introduction. On the other hand, this also leads me to question this omission as potential comparisons between reconstructed (from data), simulated, and resampled GMST calculations would be highly interesting. However, I recognize that this may be beyond the scope of this work—accordingly, I feel that the authors should preface this aspect and consider using the term metadata in their abstract and text. I think that the title of the manuscript should include/reflect 'sensitivity/uncertainty experiments' and/or 'pseudo-proxy experiments' because in its current form, without comparing simulated and actually reconstructed GMST, I do not think the authors can claim to test the 'reliability the reliability of global surface temperature reconstructions of the last glacial cycle'. Rather, they are testing the reliability of the methodologies used to create global mean reconstructions… hence I feel that a title revision is needed.**

- You are correct, we only use the metadata of the J20 dataset, not the actual measurements. See response to general comment from Bryan Lougheed. However, producing and analysing the GMST reconstruction based on the J20 data is out of scope for this manuscript.
- We argue that the pseudo-proxy experiments provide a more robust methodology to test the reliability of the reconstructions than model-data comparisons. In particular, directly comparing model outputs and proxy data suffers from the same limitations as the one we face when extrapolating our quantitative results from model data to real reconstructions (e.g. the LGM bias). However, our evaluation of the reconstruction algorithm still provides further insights, in particular regarding which timescale can be reliably reconstructed. We reformulated the introduction in that direction.

Corrected title: "*Testing the reliability of global surface temperature reconstructions of the last glacial cycle with pseudo-proxy experiments*"

**Major Comment 3**
**Inclusion of 'Full PP at random locations': It appears that the authors do not show any results from this experiment, yet it plays an important role in their analysis (see, e.g., Lines 335–340: "In addition, location resampling and latitude band configurations, which aim to account for it, are not large enough to cover the bias. Yet, the pseudo-proxy experiments using random proxy locations can reproduce the simulated MSST.") I recommend that the authors make a plot showing results from this experiment and to be more quantitative/precise regarding the ability or lack thereof of these simulations to reproduce simulated mean SSTs.**

We designed the experiment "Full PP at random locations" mostly to investigate the impact of proxy numbers on the reconstructions (L. 350-356). We realised later that they could also be used to study the effect of locations biases. The respective timeseries are shown below in Fig.S1 . We can see that the increased number of records has only a small impact on the mean reconstructed values, and that these are very close to a smoothed version of the simulated GMST. However, the increase in the number of proxy time series reduces noticeably the confidence interval.

Added Fig.S1 and references to it. (L340, 355)

**Major Comment 4**
**Attribution of a specific set of J20 locations as a significant bias: Based on the last point, the authors state that "Therefore, the bias is caused by the specific set of locations in the J20 dataset: there is an over-representation of regions with strong LGM cooling". Whereas this assessment may be accurate, I do not find the regions that the authors identify to be a convincing explanation (e.g., NW Atlantic/Kurushio extension)—instead, it seems to me that this is a bias related to the relative proximity of core locations to continents—where land-based cooling strongly impacts these sites as opposed to open-ocean marine conditions. Is it possible for the authors to combine inferences from the 'Full PP at random locations' or another sensitivity experiment to test this possibility?**

We show in Fig. S3 the map of LGM (19-23kyr) temperature anomaly as used to compute the pseudo-proxies for the 5 simulations, with the location of records with data in both the last 5kyr and the LGM. There is no clear specific cooling pattern at the proximity of the continents in the simulations. By contrast, there are large differences in and between the North Pacific and the North Atlantic, where the sampling of location is quite heterogeneous. In the text, we point in particular to undersampled regions where the cooling is weaker, hence explaining the cold bias in the reconstructions. Note that we do not rule out a cooling bias due to the proximity of the continents in the proxy reconstructions. The simulations we use simply do not show such an effect.

Added Fig. S3 and references to it (L343)

**Minor Suggestions**
**Line 81: "…needed to develop our evaluation standards." Please rephrase.**
- This sub-sentence is not needed and indeed a bit confusing. We removed it.

**Lines 91–93: Are there only 7 (112–105) unspecified datasets? Figure 1 says otherwise—please clarify.**
- No there are 112. 105(annual)+112(unspecified)+48(pseudo-annual) = 265 (total)

**Lines 162–163: Please consider adding more information to contextualize why the following steps are being performed. This would be a great spot to clarify the involvement/extent of usage (or lack thereof) of the actual records within J20.**
- What is used from J20 is more relevant for the construction of the pseudo-proxy, and we prefer to add it in the data section.
Text added: "*We decide to adapt the S16 algorithm to make full use of the J20 dataset (age ensembles, records with no data in the last 5kyr), and improve the uncertainty quantification by the algorithm.*"

**Lines 236–237: 0.26 K and 0.23 K seem to be exceedingly low values for analytical uncertainty. Does this take into account sampling uncertainty (see e.g. Thirumalai et al. 2013) which is the uncertainty that foraminiferal shells (with lifespans of a ~month) would have grown at different points of time within the sampling interval, and thus will have uncertainty in reconstructing the 'interval mean'? If sedproxy takes this into account, it would be good to mention this aspect.**
- Sampling uncertainty is applied for Mg/Ca as explained in the paragraph just above (L.228-235). The values are otherwise justified in Dolman and Laepple 2018.

**Lines 245–246: Have you considered performing a depth-sensitivity test? i.e. instead of the uppermost grid location, what about the integration of the top 50 m—which is a more realistic scenario for the proxy integration of temperature signals for the chosen sensors. Perhaps this also might explain the cool bias?**
- The main limitation to analyse the sensitivity to depth is data availability. Using a constant depth would most likely lead to slightly smaller amplitude of variation and to warmer LGM anomaly. This does not constitute an explanation to the cool bias however, but instead add a layer of uncertainty. Change in habitat depth through time can also be even more impactful, as we mention in the discussion / outlook of the paper.

**General comment on Figures: Where there are many lines depicted in figures, it is very difficult to parse the colors of each line (especially on the spectral plots) and to associate them with the legend. I would strongly consider using a different backdrop color or thicker lines with a different subplot layout to more cleary delineate results from various experiments.**
- Thank you for the feedback, we increased the thickness of the lines, particularly in the legend.

**Discussion and Lines 545–555: The authors should consider discussing the values of scaling utilized in Clark et al. (2024) and how they fit within this portion of the discussion.**
-> We present in Fig. S4 a scatter plot of GMST VS GSST in the simulation we used, including Snyder 2016 and Clark et al. 2024, with uncertainty. In short, the two are very similar, but the uncertainty range behave differently. Hence, it doesn't seem that the polynomial fit of Clark et al 2024 brings a better agreement. In addition, we would argue that neither a purely additive noise as in Clark et al 2024 or a purely multiplicative noise as in Snyder 2016 are able to capture the uncertainty correctly. Part of the issue in Clark et al. 2024 could be that they only use CESM-based simulations for the colder than PI part of the fit, which undersample the uncertainty. We will add a couple of sentences in the paragraph about the new fit from Clark et al. 2024, thank you for pointing it out.

Added Fig.S4
Added text:
L550: "*Similarly, Clark et al. (2024) uses a scaling factor, although with a different definition.*"
L 555: "*This behaviour suggests going beyond the simple LGM-to-PI ratio used to compute the scaling factor in S16, or even the one-to-one match between MSST and GMST supposed in Clark et al. (2024).*"